# Efficient Continuous-Depth Modeling with GRU Equivalents

**Ayan Banerjee** [1]  **Bin Xu** [2]  **Sandeep Gupta** [1]

## Abstract

Continuous-Depth Neural Networks (CDNNs), including Neural Ordinary Differential Equations (ODEs) and Liquid-Time-Constant Neural Networks (LTC-NN), suffer from high computational costs due to solving numerous nonlinear ODEs during training and inference. We introduce Continuous Depth Acceleration (CoDA), a framework that leverages Mori–Zwanzig/Koopman operator theory to replace continuous-depth layers requiring multiple nonlinear ODEs with a compact GRU module, a single low-dimensional linear ODE, and a dense layer. We prove PAC learnability of CoDA, establishing that this transformation preserves accuracy and can be applied repeatedly across multiple layers with unified backpropagation. Experiments on the Liquid Foundation Model (LFM-1.2B) demonstrate $6.7\times$ training speedup and $1.8\times$ inference speedup without loss of accuracy. Across six real-world LTC-NN applications, CoDA consistently outperforms state-of-the-art acceleration techniques—including neural flows, model order reduction, and variational formulations—in both training and inference time while maintaining competitive or superior accuracy. The implementation and datasets are publicly available at https://github.com/ImpactLabASU/CoDA-ICML2026.

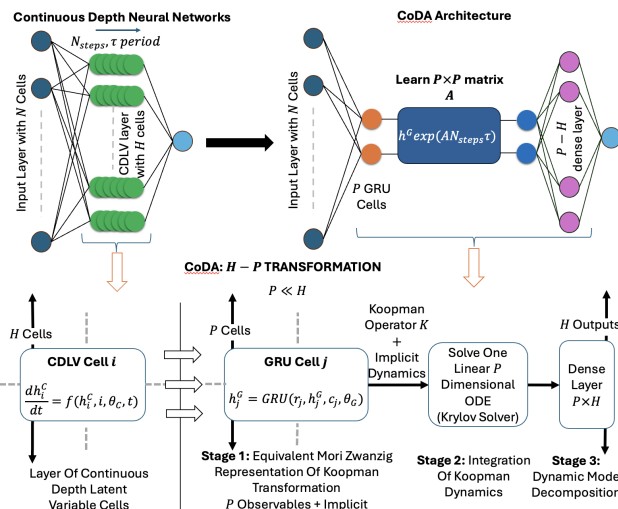

*Figure 1.* CoDA defines the $H$-$P$ transformation where any continuous-depth latent variable (CDLV) layer with $H$ cells can be replaced by a combination of $P$ GRU cells ($P \ll H$), a single $P$-dimensional linear ODE, and a $P \times H$ dense layer. **Top left:** A CDLV layer with $H$ cells, each solving a nonlinear ODE $\frac{dh_i^C}{dt} = f(h_i^C, i, \theta_C, t)$ over $N_{steps}$ and period $\tau$. **Top right:** The CoDA equivalent architecture replacing $H$ nonlinear ODEs with $P$ GRU cells that generate Koopman observables, followed by a single linear ODE solved via matrix exponentiation $h^G \exp(A N_{steps} \tau)$ using a learned $P \times P$ matrix $A$, and a dense layer that recovers the original $H$-dimensional outputs. **Bottom:** The three-stage $H$-$P$ transformation pipeline: (1) Mori-Zwanzig/Koopman transformation producing $P$ observables, (2) integration of Koopman dynamics via Krylov solver, and (3) dynamic mode decomposition to recover $H$ outputs.

## 1. Introduction

Continuous-depth neural networks (CDNNs) parameterize hidden-state evolution via learned differential equations and have been applied across diverse domains including human action recognition (Rubanova et al., 2019), gesture

[1]School of Computing and AI, Arizona State University, Tempe, USA [2]School of Electrical, Computer and Energy Engineering, Arizona State University, Tempe, USA. Correspondence to: Ayan Banerjee <Ayan.Banerjee@asu.edu>, Bin Xu <binxu4@asu.edu>, Sandeep Gupta <Sandeep.Gupta@asu.edu>.

*Proceedings of the 43rd International Conference on Machine Learning*, Seoul, South Korea. PMLR 306, 2026. Copyright 2026 by the author(s).

recognition (Wagner et al., 2014), and also foundational models (Liquid AI, 2025). Neural Ordinary Differential Equations (NODEs) underpin continuous-depth residual networks and continuous normalizing flows in generative modeling (Chen et al., 2018), which is extended to Liquid-Time-Constant Neural Networks (LTC-NN) that achieve state-of-the-art (SOTA) sequence modeling by modulating input driven nonlinear dynamical systems (Hasani et al., 2021). Recently, Liquid Foundation Models have been introduced as large-scale, general-purpose AI systems for video, audio, text, and time-series data (Liquid AI, 2024; 2025). Despite their expressivity, such models rely on iterative ODE solvers both during training and inference time, introducing substantial computational overhead, latency, and energy consumption.

Acceleration of neural networks (NNs) has garnered significant recent interest, with approaches broadly classified into three orthogonal categories that can be used in conjunction:

**a) Hardware acceleration:** Early approaches implement entire NN architectures on hardware accelerators such as FPGAs or analog circuits. However, these techniques have not been extensively explored for CDNNs, as hardware acceleration of ODE solvers either introduces quantization errors or requires substantial hardware resources, such as lookup table (LUT) usage in FPGAs (Damaj et al., 2024).

**b) Loss function modulation:** Zhao et al. (2025) propose a novel variational loss function that bypasses ODE integration and can be integrated into any NODE-based architecture. However, this approach has not been validated on other CDNNs such as LTC-NN.

**c) Alternate network architecture:** There are two types of approaches for altering the network architecture for enabling acceleration: *i) accuracy sacrificing approximate network structures*, there are three subtypes: 1) quantization of the original network such as logic gate networks, quantized NNs, binary NNs, which have not been explored for CDNNs, or 2) sparsification of the original model such as sparse NNs reducing the number of weights to be trained (not explored in CDNNs), or 3) recently proposed model order reduction (Lehtimäki et al., 2022) for NODE, where the test time acceleration is achieved by eliminating NODE outputs that have low Eigen values as evaluated through single value decomposition. *ii) accuracy preserving equivalent architecture*, where a CDNN layer can be replaced by the equivalent layer which has a learnability guarantee with no loss of accuracy. Neural flows (Biloš et al., 2021) is an architecture equivalent to NODE, which bypasses ODE solvers and directly models the integration of the NODE forward pass using Gated Recurrent Units (GRU) flows. While neural flows achieve good acceleration without loss of accuracy, it is only defined for NODE based CDNN. In this paper, we define an equivalent computationally efficient architecture for CDNNs in general, **Co**ntinuous **D**epth **A**cceleration (CoDA), that can replicate hidden states of the continuous depth latent variable (CDLV) layer without loss of accuracy. CoDA is generally applicable to any CDNN including NODE, LTC-NN and Continuous Time Recurrent Neural Networks (CT-RNN).

**Positioning relative to prior work:** CoDA is the *only* accuracy-preserving, *training-time-reducing*, architecture-level transformation applicable across NODE, LTC-NN, and general CDLVs. Neural flows (Biloš et al., 2021) address only NODE and do not reduce training time. Model order reduction (Lehtimäki et al., 2022) accelerates inference but not training and sacrifices accuracy. Variational formulations (Zhao et al., 2025) modify the loss function but remain limited to NODE. Sparse/contextual methods (e.g.,

DejaVu (Liu et al., 2023)) exploit activation sparsity but do not address the fundamental ODE solver bottleneck.

**Main Contributions:** Our work makes two key contributions:

**(1) Theoretical Contribution:** We leverage Koopman theory (Brunton et al., 2022) and its Mori–Zwanzig formalism (Lin et al., 2021) to prove any $H$-cell CDLV layer can be replaced by an equivalent $H$-$P$ transformation (Figure 1): a $P \ll H$ GRU module providing finite-dimensional Koopman observables, a single $P$-dimensional linear ODE solver, and a dense layer recovering the original $H$-dimensional dynamics via dynamic mode decomposition (Kaiser et al., 2018). We establish PAC learnability by showing the transformation preserves Rademacher complexity.

**(2) Systems Contribution:** CoDA is explicitly a systems-for-ML contribution. At the operator level, it replaces iterative nonlinear ODE integration with matrix exponentiation, enabling kernel fusion, deterministic execution, and hardware-friendly Krylov solvers. For compute and memory efficiency, we provide full FLOP, I/O, cache, and energy breakdowns (Tables 5, 2), demonstrating up to $6.7\times$ training speedup, $1.8\times$ inference speedup, and $2$–$90\times$ energy savings with reduced memory traffic. For hardware mapping, linear ODEs combined with ConvGRUs map naturally to tensor cores, systolic arrays, and streaming pipelines on various hardware accelerators.

**Theoretical Novelty:** We show that $H - P$ transformation of a CDLV layer in a CDNN, has the same Rademacher number as the forward pass of the original CDLV layer hence the probably approximately correct (PAC) learnability guarantees can be extended to CoDA.

**Why CoDA accelerates?** The CoDA architecture bypasses the requirement to backpropagate through $H$ nonlinear ODEs and replaces them with a single smaller $P$ dimensional linear ODE that can be solved with Krylov solvers (Liesen & Strakos, 2013) using matrix multiplication. This is the key step towards acceleration and energy savings. Furthermore, CoDA uses GRU that strikes a fine balance between: a) the capability to perform **convolution operation**, a key requirement for equivalence (Theorem F.1), which is not available in vanilla RNN, and b) **speed**, GRU has far fewer weights than Long Short-Term Memory (LSTM) cells yet having the capacity to retain memory over long sequences.

## 2. Preliminaries

Assume an $H$-dimensional nonlinear system defined by

$$\frac{dh}{dt} = f\big(h(t), i(t), \theta, t\big), \tag{1}$$

where $h(t) = [h_1(t), \ldots, h_H(t)]^\top \in \mathbb{R}^H$ is the state vector, $i(t)$ denotes the $N$-dimensional input, $\theta$ is a parameter matrix, and $f$ is a Lipschitz continuous, integrable function.

**Implicit and Measured Dynamics.** We partition $h(t)$ into *measured dynamics*, whose components are directly observable, and *implicit dynamics*, whose components are unobserved but influence the measured variables. We assume an observable system in which all state variables affect the measurements, although not all are measured.

**ODE Solvers.** ODE solvers fall into two categories: nonlinear and linear. Nonlinear solvers, such as ODE45 (Senan et al., 2007) or Runge–Kutta methods (Cartwright & Piro, 1992), require iterative integration and are computationally intensive. Linear solvers exploit matrix exponentiation and Krylov-subspace methods for large systems, offering significantly lower computational cost (Liesen & Strakos, 2013).

**Koopman Theory.** Koopman theory (Brunton et al., 2022) represents the evolution of a nonlinear system via an (infinite-dimensional) linear operator. Given a measurement function $\mathcal{G} : \mathbb{R}^H \to \mathbb{R}^P$, the Koopman operator $\mathcal{K}$ satisfies

$$\mathcal{K}\mathcal{G}\big(h(t), i(t)\big) = \mathcal{G}\big(f\big(h(t), i(t), \theta, t\big), i(t)\big), \qquad (2)$$

with $P \to \infty$ in the general case.

**Mori–Zwanzig Formalism.** In practice, the Koopman representation is sparse in the observable manifold. Absent explicit sparsity, one can combine Koopman theory with its Mori–Zwanzig formalism (Lin et al., 2021), yielding a decomposition of the observables into measured components $\mathcal{G}_M$ and implicit components $\mathcal{G}_I$:

$$\begin{bmatrix} \dot{\mathcal{G}}_M \\ \dot{\mathcal{G}}_I \end{bmatrix} = \begin{bmatrix} \mathcal{K}_M & \mathcal{K}_{MI} \\ \mathcal{K}_{IM} & \mathcal{K}_I \end{bmatrix} \begin{bmatrix} \mathcal{G}_M \\ \mathcal{G}_I \end{bmatrix}. \qquad (3)$$

Here, $\mathcal{K}_M$ captures the observable dynamics, $\mathcal{K}_I$ the implicit dynamics, and $\mathcal{K}_{IM} = \mathcal{K}_{MI}$ the interactions between the two.

**CDNN with Continuous Depth Latent Variables.** The general CDNN architecture (Figure 1) employs continuous depth latent variable (CDLV) nodes, such as NODE or LTC-NN, to model dynamical systems. Training minimizes a task-specific loss (e.g., cross-entropy) often combined with a physics-based loss that enforces known physical laws.

In this work, each CDLV node (Chen et al., 2018) has a forward pass given by Equation 4 over a time horizon $\tau$:

$$h_i^C(t) = h_i^C(t - \tau) + \int_{t-\tau}^t f\big(h_i^C(s), i(s), \theta_i, s\big)\, ds, \quad (4)$$

where $h_i^C(t)$ denotes the $i$th hidden state. Specific examples of $f$ include $f(h_i^C, i, \theta_i, t) = \tanh\big(W_i[i(t), h_i^C(t)]^\top + b_i\big)$ for NODE architecture (Chen et al., 2018), and $\theta_i = \{W_i, b_i\}$.

**General Continuous-Depth Cells.** Our theoretical framework applies to any continuous-depth network satisfying Lipschitz continuity and integrability, including LTC-NN and continuous-time recurrent neural networks (Hasani et al., 2021).

**Gated Recurrent Unit.** A GRU cell can be interpreted as an efficient discretization of an ODE integration, employing reset and update gates. Its forward pass is

$$h_i^G(t) = r_i(t-1) \circ h_i^G(t-1) + \big(1 - r_i(t-1)\big) \circ c_i(t-1), \qquad (5)$$

$$r_i(t) = \sigma\big(W_r i(t-1) + U_r h_i^G(t-1) + b_r\big), \qquad (6)$$

$$c_i(t) = \sigma\big(W_h i(t-1) + U_h(r_i(t-1) \circ h_i^G(t-1)) + b_h\big), \qquad (7)$$

where $\circ$ denotes the convolution operation, and $\sigma(\cdot)$ is an activation function. The GRU can have different activation functions for the reset and update gates.

**Probably Approximately Correct (PAC)** PAC theory provides statistical equivalence in accuracy for machines of equal expressivity (Haussler & Warmuth, 2018) measured by Rademacher number.

## 3. Theoretical foundation of CoDA

Under the Lipschitz continuous assumption of the forward pass of CDLV cells, utilizing Lemma F.1, we can decompose the forward pass into an observable dynamics through the hidden layer outputs and implicit dynamics that is a result of unmodeled components.

**Lemma 3.1** (CDLV Decomposition). *The representation output of a CDLV layer with $H$ cells can be decomposed into a linear combination of a dynamical system that is linear on a set of observable nonlinear library functions and a nonlinear convolution of the library functions.*

**Proof sketch:** *Let us assume that $h^C(t)$ denotes the vectorized form of Equation 4 where $h_i^C(t)$ is one element of the vector. The MZ formalism in Equation 3 applies to $f$. Integrating Equation 3 using Laplace transform and assuming long sequences of data, we can obtain the following form for the observable dynamics $\mathcal{G}_M$ (full proof in supplement).*

$$\dot{\mathcal{G}}_M = \underbrace{\mathcal{K}_M \mathcal{G}_M}_{\text{Part 1: Linear}} + \underbrace{\mathcal{K}_{MI} \int_{-\infty}^t e^{t-s\mathcal{K}_I} \mathcal{G}_M(s)\, ds}_{\text{Part 2: Implicit dynamics convolution}} \qquad (8)$$

*After evolving for $\mathcal{G}_M$ following Equation 8, the hidden layer output $h^C(t)$ can be recovered as $h^C(t) = \mathcal{K}_M \mathcal{G}_M(t)$. Part*

*1 of Equation 8 is a linear dynamical system on $\mathcal{G}_M$, and part 2 is a nonlinear function that computes convolution of $\mathcal{G}_M$ and an exponential decay function. $K_{MI}$ is unknown Koopman implicit interaction matrix.*

Part 1 being a linear dynamics can be directly modeled by GRU forward pass if reset and update gates have only non-zero bias and other weights are defaulted to 0. However, it is not intuitive that Part 2 can be modeled using GRU gates. To establish this we prove Theorem F.1, which also defines the equivalent architecture.

**Theorem 3.1.** *$\exists W_r$, $U_r$, $b_r$, $W_h$, $U_h$, and $b_h$ such that for any $\epsilon > 0$ and $0 < \alpha < 1$, GRU layer forward pass $h^G(t)$ can estimate implicit dynamics in the MZ formulation of CDLV forward pass $h^C(t)$ with accuracy $\epsilon$, with UAT guarantee for long sequences and PAC guarantees for all sequences, $Pr(||h^{\dot G}(t) - \dot{\mathcal{G}} - \mathcal{K}_M \mathcal{G}_M|| < \epsilon) > (1-\alpha)$.*

**Proof sketch:** *We prove the theorem by constructing the GRU layer that accurately estimates the implicit dynamics part of Equation 8.*

**Step 1, modeling time:** *We first add a GRU cell $k$ that models time. This means that $W_r(i,k) = 0, U_r(i,k) = 0, W_h(i,k) = 0, U_h(i,k) = 0 \forall i$. Then we have $b_r(k) = b_h(k) = B_k$, where $B_k$ is a large positive value. This forces the output $h_k^G(t) = h_k^G(t-1) + c$, where $c = tanh(B_k) \approx 1$, making it keep time.*

**Step 2, $P$ GRU cells:** *We now add $P$ GRU cells with the following weight configuration:*

**Reset Function** *Let for each cell $i$ except for the time cell $k$, $\forall i \neq k$, we have a reset weight $W_r(j,i) = 0, \forall j, U_r(j,i) = 0, \forall j \neq k, U_r(k,i) = u_{ki}, b_r(i) = 0, \forall i \neq k$, and a sigmoid activation function.*

**Update Function** *For each cell $i \neq k$, let $b_h(i) = 0, \forall j \in \{1 \ldots P\}$ with learned weights $W_r$ and $U_r$.*

*The forward pass of this layer in the continuous form can be expressed by Equation 9 (full derivation in supplement).*

$$\frac{dh_i^G(t)}{dt} = \underbrace{\int_0^t e^{-(u_{ki}+B_k)(t-z)} A[h_i^G(z), i(z)]^T dz}_{\text{Part 1: Input Dependent Implicit Interaction}} \quad (9)$$

$$+ \underbrace{\sum_{j=1}^{P} U_h(j,i) \int_0^t \frac{e^{(u_{ki}+B_k)(t-z)}}{2(u_{ki}+B_k)} h_i^G(z) dz}_{\text{Part II: Input Independent Implicit Interaction}}.$$

*where $A$ is a matrix of size $(P+N) \times (P+N)$ by concatenating an identity matrix of size $P \times P$ in the upper diagonal and the matrix $W_h$ of size $N \times N$ in the lower*

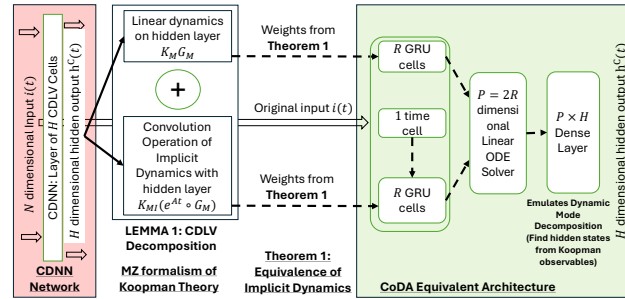

*Figure 2.* Conversion of CDLV layer in a CDNN into a CoDA block.

*diagonal.*

*We observe that Equation 8 and 9 are of the same class of functions i.e. a nonlinear polynomial function convolved with an exponential function. Hence both the implicit dynamics interaction part of forward pass of CDLV cell in Equation 8 and forward pass of GRU Equation 9 have the same Rademacher number (Gnecco, 2008). Hence Equation 9 has PAC learning guarantee for learnability of the implicit dynamics interaction part. This implies that $\exists \epsilon > 0, 0 < \alpha < 1$ such that $Pr(||h^{\dot G}(t) - \dot{\mathcal{G}} - \mathcal{K}_M \mathcal{G}_M|| < \epsilon) > (1-\alpha)$.*

| Weight | Block 1 Constraint | Block 1 weights | Block 2 Constraint | Block 2 weights |
|---|---|---|---|---|
| $W_r$ | all entries | 0 | all entries | 0 |
| $U_r$ | all entries | 0 | $U_r(j,i), j \neq k$ | 0 |
| | | | $j = k$ | $u_{k,i}^{(2)}$ |
| $b_r$ | all entries | *learned* $B_i^{(1)}$ | $i \neq k$ | 0 |
| | | | $i = k$ | $B_k^{(2)}$ |
| $W_h$ | all entries | learned | all entries | learned |
| $U_h$ | all entries | 0 | all entries | learned |
| $b_h$ | all entries | *learned* $b_i^{(1)}$ | all entries | 0 |

*Table 1.* $P + 1 = 2R + 1$–GRU weight configurations based on Theorem F.1

### 3.1. CoDA Layer Architecture

Lemma F.1 and Theorem F.1 define the CoDA equivalent architecture to a layer of CDLV cells, as illustrated in Figure 2. The $N$-dimensional input $i(t)$ is fed directly into a GRU layer with $P + 1 = 2R + 1$ cells, organized into three functional blocks. Block 1, consisting of $R$ cells, computes the $R$-dimensional linear dynamics corresponding to Part 1 in Equation 8. Block 2, also with $R$ cells, computes Part 2 in Equation 8. A single additional cell models the temporal component. The weight configuration for each cell follows from Theorem F.1 and is summarized in Table 1.

The outputs from Blocks 1 and 2 are concatenated to form a $P$-dimensional vector, which serves as the Koopman observables in its Mori–Zwanzig formulation. This $P$-dimensional output is then integrated using a Krylov subspace method with matrix exponentiation, employing a learned $P \times P$

matrix $A$ over total integration time $t = N_s\tau$. The solution to the resulting linear ODE is $h^G(t)\exp(AN_s\tau)$, where the matrix exponential is computed via Padé approximation (Arioli et al., 1996). Finally, the output is passed through a $P \times H$ dense layer that learns the mapping $\mathcal{K}_M$, recovering $h^C(t)$ according to Dynamic Mode Decomposition (DMD) theory (Schmid, 2022).

```python
class NODE(nn.Module):
  def __init__(self, H):
    self.odefunc = nn.Sequential(
      nn.Linear(H, H), nn.Tanh())

  def forward(self, x):
    # Iterative ODE solving (slow)
    for step in range(N_steps):
      x = x + dt * self.odefunc(x)
    return x
```

```python
class CoDA(nn.Module):
  def __init__(self, H, P):  # P << H
    self.gru = nn.GRU(H, P, batch_first=True)
    self.A = nn.Parameter(torch.randn(P, P))
    self.dense = nn.Linear(P, H)

  def forward(self, x):
    z, _ = self.gru(x)
    Phi = torch.matrix_exp(self.A) # Koopman obs
    out = z @ Phi.T        # linear ODE solution
    return self.dense(out)  # recover H-dim
```

*Figure 3.* Comparison of NODE and CoDA implementations. **Left:** Original NODE requires iterative ODE solving with $N_{steps}$ sequential operations. **Right:** CoDA replaces iteration with a single matrix exponentiation, reducing $O(N_{steps} \cdot H^2)$ to $O(P^2)$ where $P \ll H$.

Figure 3 illustrates how CoDA serves as a drop-in replacement for existing NODE implementations. The original NODE layer solves an ODE with $H$ hidden units via iterative numerical integration, requiring $N_{\text{steps}}$ sequential evaluations of a nonlinear `odefunc` with cost $O(N_{\text{steps}} \cdot H^2)$. This sequential dependency prevents parallel execution across the time axis and causes repeated memory accesses at each integration step, leading to poor data reuse and memory-bandwidth bottlenecks. In contrast, the CoDA layer replaces this with the $H$-$P$ transformation, implemented as three lightweight components: (1) a GRU module that maps the $H$-dimensional input into $P$-dimensional Koopman observables $z$ with $P \ll H$, (2) a learned $P \times P$ parameter matrix `self.A` whose matrix exponential $\Phi = \exp(AN_s\tau)$, computed once per forward pass via `torch.matrix_exp`, propagates the linear Koopman dynamics, and (3) a dense layer that projects $z\Phi^\top$ back to $H$ dimensions via dynamic mode decomposition. The key computational advantage is replacing $N_{\text{steps}}$ sequential nonlinear updates with a single matrix exponentiation plus a batched linear projection: per-token cost drops from $O(N_{\text{steps}} \cdot H^2)$ to $O(HP + P^2 + PH)$,

dominated by the two $H$–$P$ projections since $P \ll H$. Eliminating the sequential dependency also enables full parallelization across the time and batch axes and improves hardware utilization, while the compact $P$-dimensional intermediate representation reduces memory footprint and improves cache locality. Together these optimizations make CoDA readily integrable into existing frameworks while delivering significant acceleration.

# 4. Time and Energy Efficiency of CoDA

In this section, we provide a mathematical analysis of CoDA's time and energy efficiency compared to standard CDLV layers.

## 4.1. Time Efficiency

We evaluate computational complexity by counting scalar multiplications for an $H$-cell CDLV layer versus the $H$-$P$ transformation in CoDA.

**CDLV Layer Complexity:** For an $H$-cell NODE layer with input size $N$, each evaluation of the ODE function $f$ requires: (1) $N \times H$ multiplications for the weight matrix, and (2) $2Hd$ multiplications for computing tanh activation via Padé approximation of degree $d$. Thus, one evaluation of $f$ costs $NH + 2Hd$ multiplications. For an ODE solver with $N_s$ steps, function $f$ is evaluated $N_s$ times. Additionally, each step $i$ requires a weighted sum of the previous $i - 1$ steps, contributing $HN_s(N_s - 1)/2$ multiplications. The total complexity for an $H$-cell NODE layer is therefore:

$$C_{\text{NODE}} = N_s NH + 2N_s Hd + \frac{HN_s(N_s - 1)}{2} \quad (10)$$

**CoDA Complexity:** The $H$-$P$ transformation consists of three components: (1) a GRU module requiring $3NP + 3P^2 + 3P$ multiplications, (2) matrix exponentiation via Padé approximation of degree $d$ requiring $dP^3$ multiplications, and (3) a dense layer requiring $P \times H$ multiplications. The total complexity is:

$$C_{\text{CoDA}} = 3NP + 3P^2 + 3P + dP^3 + PH \quad (11)$$

**Speedup Analysis:** The theoretical speedup $\sigma_{\text{CoDA}}$ of CoDA over NODE is:

$$\sigma_{\text{CoDA}}(\text{NODE}) = \frac{N_s NH + 2N_s Hd + HN_s(N_s - 1)/2}{3NP + 3P^2 + 3P + dP^3 + PH} \quad (12)$$

A similar analysis applies to LTC-NN (see supplement), with modifications to the numerator. This estimate is conservative, as it does not account for optimized kernels available for matrix exponentiation that are unavailable for iterative ODE solvers.

Two key observations emerge from Equation 12. First, for fixed $H$, speedup decreases as $P$ increases; however, in practice, $P < \sqrt{H}$ typically achieves comparable or superior accuracy, ensuring substantial speedup. Second, speedup scales as $O(N_s^2)$ with increasing ODE solver steps $N_s$, making CoDA particularly advantageous for high temporal resolution (deeper) CDNNs where greater accuracy is required.

## 4.2. Energy Efficiency

Energy consumption corresponds to the area under the power curve and is dominated by peak power during the most computationally intensive operations. For CDLV layers, peak power occurs during iterative ODE integration, which involves repeated sequential memory accesses. For CoDA, peak power is determined by matrix exponentiation, which benefits from highly optimized parallel implementations with superior memory access patterns, resulting in lower overall energy consumption.

## 5. Evaluation and Results

Our main goals of evaluation are:

**a) Large-scale analysis and ablation study:** We evaluate CoDA on the Liquid Foundational Model (LFM-2) (Liquid AI, 2025), a hybrid liquid architecture combining multiplicative gating, short-range convolutions, and grouped query attention. The student model (LFM2-1.2B) with $H = 1024$-cell LTC-NN layers is accelerated by replacing each layer with a $P = 16$ GRU-based CoDA block, while LFM2-7B serves as the teacher.

*Dataset:* We use the WildChat dataset (Feuer & Hegde, 2025) with 650K real-world ChatGPT dialogues and AID Embodied Prompt. We train on 80% and test on 20%.

**b) Generality assessment:** We test CoDA on six different types of embodied applications that integrate time-series tasks with textual prompts.

*Model:* For each dataset, we compare LTC-NN layers with $H = 128$ against CoDA with $P = 16$.

*Comparators:* We include GRU flows (GRUF) (Biloš et al., 2021), NODE, LTC-NN, and Long Short-Term Memory (LSTM). For brevity in tables we write **LTC** for LTC-NN.

**c) SOTA comparison:** For the AID embodied application we compare CoDA with two state-of-the-art CDLV acceleration methods: NODE + model order reduction (NODE-MOR) (Lehtimäki et al., 2022), and NODE + variational formulation (NODE-VF) (Zhao et al., 2025).

*Model/Dataset:* For NODE-MOR, we use MNIST with a NODE layer (512 cells) and MOR values of 100, 200, and 500, comparing against an $H \rightarrow P$ CoDA block with $P = 16$. For NODE-VF, we use the synthetic dataset from (Zhao

et al., 2025), replacing NODE layers with CoDA blocks of $P \in [16, 1024]$.

## 5.1. Platform and measurement method

To evaluate the performance of CoDA, experiments were conducted across three hardware platforms: (1) a server system with an Intel Xeon w9-3475X CPU and an NVIDIA RTX 6000 GPU (48 GB memory), (2) an NVIDIA Jetson Orin Nano as the mobile GPU platform, and (3) a Xilinx Alveo U280 FPGA board. The model was implemented using PyTorch 2.4 for GPU and mobile GPU platforms. Power consumption was measured using an external power meter for accurate readings across all platforms.

**Statistical analysis:** We provide standard deviation of all results in supplement.

*Table 2.* Ablation study comparing the impact of different architectural variants on training accuracy, energy efficiency, training time, and inference performance. `T` is Timing, `Ep` is Epoch. The best result per row is shown in **bold**, and the second-best is underlined.

| Model | Train Loss | T/Ep (s) | Train T (s) | Energy/E (J) | Test Loss | Test T (s) |
|---|---|---|---|---|---|---|
| LTC | 0.0970 | 0.125 | 3815.64 | 8.000 | 4.1308 | 34.65 |
| CoDA − cuda | **0.0834** | 0.0835 | 829.93 | 5.43 | 4.247 | 19.33 |
| **CoDA** | 0.0875 | **0.028** | **567.11** | **1.792** | 4.0191 | 19.24 |
| CoDA-LODE | 0.0889 | 0.050 | 1200.93 | 3.200 | 4.1398 | 19.10 |
| CoDA 1024 | 0.0897 | 0.0714 | 676.21 | 4.5696 | 4.0900 | 19.39 |
| CoDA+sparse | 0.2519 | 0.100 | 1892.58 | 6.400 | 3.9560 | 19.29 |
| CoDA-LODE+sparse | 0.3391 | 0.0769 | 1844.78 | 4.9216 | **3.9031** | **19.02** |

## 5.2. Results

**a) Large-scale analysis and ablation study:** Table 2 shows that CoDA achieves a $6.7 \pm 1.1\times$ speedup in training and a $1.8 \pm 0.3\times$ speedup in testing for the LFM2-1.2B model. The training and testing losses of CoDA are statistically equivalent to the original LFM2-1.2B model, with CoDA exhibiting lower mean loss, demonstrating architectural equivalence. CoDA is also the most energy-efficient variant, consuming only 1.7 J per epoch—$4.5\times$ less than the original LFM2-1.2B model.

Ablation analysis reveals that removing the linear ODE component (CoDA-LODE) increases both training and test loss while drastically increasing total training time, indicating that the linear ODE plays a critical role in maintaining CoDA's equivalence with LFM2-1.2B. Replacing the dense layer with 1024 GRU cells (CoDA 1024) degrades all metrics. Interestingly, applying sparse NN acceleration to the dense layer (CoDA + sparse) improves test loss but has minimal effect on test time. Additionally, disabling CUDA (CoDA − cuda) increases both time per epoch and energy consumption compared to the CUDA-enabled CoDA.

Figure 4 shows that for the smallest $P$ the training time is low while the test loss is high. As $P$ increases, CoDA be-

*Table 3.* Comparison of model performance across six applications during inference, including inference time per epoch, energy consumption, and inference loss. The best result per row is shown in **bold**, and the second-best is underlined.

| Application | Inference Time / Epoch (s) | | | | | Energy / Epoch (J) | | | | | Testing Loss | | | | |
|---|---|---|---|---|---|---|---|---|---|---|---|---|---|---|---|
| | **CoDA** | GRUF | NODE | LTC | LSTM | **CoDA** | GRUF | NODE | LTC | LSTM | **CoDA** | GRUF | NODE | LTC | LSTM |
| occupancy | **0.31** | 0.76 | 27.15 | 0.92 | 0.51 | **24.18** | 57.76 | 2172.00 | 77.28 | 39.78 | **0.13** | 0.25 | 0.25 | 0.31 | 0.25 |
| ozone | **0.03** | 0.04 | 2.57 | 1.21 | 0.04 | **2.70** | 3.72 | 277.56 | 134.31 | 4.08 | 0.13 | 0.13 | 0.24 | 0.13 | **0.12** |
| person | **0.03** | 0.58 | 1.01 | 2.50 | 0.04 | **2.43** | 53.36 | 112.11 | 217.5 | 4.04 | **0.45** | **0.45** | 0.80 | 0.46 | 0.55 |
| power | 0.16 | 0.07 | 0.36 | **0.001** | 0.09 | 13.6 | 5.88 | 41.04 | **0.078** | 7.02 | 0.03 | **0.0022** | 0.01 | 0.009 | 0.01 |
| har | **0.02** | 0.03 | 0.66 | 0.09 | 0.13 | **1.60** | 2.34 | 56.10 | 7.38 | 10.92 | **0.11** | 0.21 | 0.18 | 0.15 | 0.18 |
| gesture | **0.0085** | 0.01 | 0.19 | 0.21 | 0.01 | **0.6545** | 0.79 | 16.72 | 16.59 | 0.85 | **0.95** | **0.95** | 0.98 | 1.15 | 1.39 |

*Table 4.* Comparison of model performance across six applications during training, including training time per epoch, energy consumption, and training loss. The best result per row is shown in **bold**, and the second-best is underlined.

| Application | Training Time / Epoch (s) | | | | | Energy / Epoch (J) | | | | | Training Loss | | | | |
|---|---|---|---|---|---|---|---|---|---|---|---|---|---|---|---|
| | **CoDA** | GRUF | NODE | LTC | LSTM | **CoDA** | GRUF | NODE | LTC | LSTM | **CoDA** | GRUF | NODE | LTC | LSTM |
| occupancy | **0.5** | 0.62 | 20.9 | 18.42 | 0.65 | **39** | 47.12 | 1672 | 1547.28 | 50.7 | **0.0096** | **0.0096** | 0.03 | **0.0096** | 0.01 |
| ozone | **0.1** | 0.2 | 2.62 | 2.57 | 0.2 | **9** | 18.6 | 282.96 | 285.27 | 20.4 | **0.13** | **0.13** | 0.24 | **0.13** | **0.13** |
| person | **0.2** | 0.3 | 10.3 | 10.99 | 0.38 | **16.2** | 27.6 | 1143.3 | 956.13 | 38.38 | **0.37** | 0.39 | 0.79 | 0.45 | 0.5 |
| power | **4** | 5.5 | 272.6 | 210.09 | 5.7 | **340** | 462 | 31076.4 | 16387.02 | 444.6 | **0.0013** | **0.0013** | 0.007 | 0.009 | **0.0013** |
| har | **0.7** | 0.8 | 20.8 | 24.89 | 0.865 | **56** | 62.4 | 1768 | 2040.98 | 72.66 | 0.0756 | 0.082 | 0.0883 | **0.0334** | 0.097 |
| gesture | **0.1** | 0.143 | 3.25 | 2.51 | 0.152 | **7.7** | 11.297 | 286 | 198.29 | 12.92 | 0.46 | 0.5 | 0.92 | 0.96 | **0.44** |

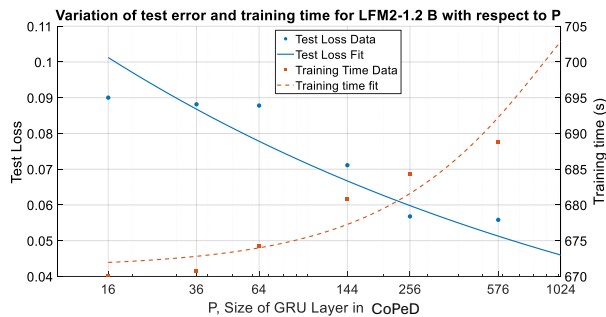

*Figure 4.* Variation of test loss and training time in CoDA acceleration of LFM2-1.2 B as GRU layer size $P$ is changed.

*Table 5.* Theoretical breakdown for Attention block versus LTC-NN block in one transformer layer when generating one token on 3 A6000-45GB GPUs.

| Component | GFLOPs | I/O (GB) | Comp. Lat. (ms) | I/O Lat. (ms) |
|---|---|---|---|---|
| Attention Block | 23.14 | 0.003588 | 0.5 | 1.73 |
| LTC-NN Block | 95.66 | 0.030565 | 1.5 | 1.75 |
| CoDA Block | 10.53 | 0.012346 | 0.17 | 0.7 |

comes more and more accurate; however, the training time increases drastically. This is expected because theoretically as $P$ increases and becomes closer to 1024 it explicitly models all dynamics in the data according to MZ formulation, which increases its chances to learn accurately.

Table 5 shows that the LTC-NN block requires substantially more compute and data movement than the Attention block per token (95.66 vs. 23.14 GFLOPs; 0.030565 vs. 0.003588 GB), which translates into a markedly higher compute latency (1.5 ms vs. 0.5 ms). In contrast, the proposed CoDA block cuts both FLOPs and I/O (10.53 GFLOPs; 0.012 GB), yielding much lower compute and I/O latencies (0.17 ms and 0.70 ms, respectively).

**b) Generality assessment:** Table 4 evaluates model performance across six applications in terms of **training** time, energy consumption, and training loss. CoDA consistently achieves a favorable trade-off across all three metrics. It

significantly reduces training energy compared to NODE and LTC-NN—for example, in the power task, CoDA uses only 340 J versus NODE's 31,076.4 J and LTC-NN's 16,387.02 J—while also achieving the lowest training loss (0.0013) among all models. GRUF performs competitively in energy and training time but occasionally incurs slightly higher loss. LSTM typically achieves the shortest training time (e.g., gesture: 0.152 s), but often at the cost of higher training losses. NODE and LTC-NN, while sometimes more accurate (e.g., har task), demand substantially more training energy and longer training times.

Table 3 compares **inference** time, energy consumption, and test loss. CoDA demonstrates a consistently strong balance between inference efficiency and predictive performance. For instance, in the power task, CoDA achieves a testing loss of 0.03 in just 0.16 s, compared to NODE's lower loss of 0.01 but with higher latency (0.36 s) and greater energy consumption. Across all applications, CoDA generally maintains inference time below 0.2 s while achieving competitive or superior accuracy. GRUF offers faster inference in some cases (e.g., gesture: 0.01 s), but at the expense of higher testing loss (0.95). Similarly, LSTM achieves fast inference but suffers from elevated losses in most tasks (e.g.,

*Table 6.* Comparison across platforms for LTC-NN and CoDA workloads on the AID dataset. M-GPU means the Mobile GPU platform.

| Workload | Avg. Error | | | RunTime (s) | | | Power (W) | | | DRAM (MB) | | | Freq (MHz) | | |
|---|---|---|---|---|---|---|---|---|---|---|---|---|---|---|---|
| | FPGA | M-GPU | GPU | FPGA | M-GPU | GPU | FPGA | M-GPU | GPU | FPGA | M-GPU | GPU | FPGA | M-GPU | GPU |
| LTC-NN Block | 3.76 | 3.993 | 2.55 | 1755.04 | 1100.82 | 1152.92 | 5.10 | 6.855 | 81 | 494.62 | 2532.45 | 6465.07 | 177 | 306 | 1410 |
| CoDA Block | 4.60 | 3.07 | 2.89 | 251.97 | 423.75 | 380.21 | 4.90 | 5.53 | 72 | 214.23 | 2355.13 | 6118.36 | 173 | 306 | 1410 |

gesture: 1.39). In contrast, NODE and LTC-NN remain energy- and time-intensive during inference, limiting their practical deployability. These results demonstrate CoDA's effectiveness in delivering generalizable performance with moderate computational overhead and low latency.

We compare CoDA against LTC-NN across different hardware platforms. Table 6 highlights the advantages of CoDA on the AID application. For accuracy, GPUs achieve the lowest average error for both workloads (2.55 for LTC-NN, 2.89 for CoDA), while FPGAs and M-GPUs perform slightly worse but remain comparable. For **runtime**, CoDA achieves a $7\times$ speedup on FPGA (251.97 s) compared to LTC-NN (1755.04 s), indicating that its compute pattern maps efficiently to FPGA pipelines. For **power efficiency**, CoDA on FPGA consumes only 4.90 W, comparable to LTC-NN (5.10 W), and both are significantly lower than GPU power consumption (72–81 W). For **DRAM footprint**, CoDA requires 214.23 MB on FPGA—less than half of LTC-NN (494.62 MB)—and far smaller than the multi-GB usage on GPUs (6118–6465 MB), making it suitable for deployment on memory-constrained edge devices.

**c) Comparison with SOTA:** Table 7 shows that CoDA ($H = 512$, $P = 16$) achieves comparable test accuracy (92%) to NODE-512-MOR (500) at 93%, while significantly reducing test time from $0.09s \pm 0.03$ to $0.03s \pm 0.01$—a $3\times$ speedup. CoDA also reduces training time per epoch from $17.4s \pm 1.3$ to $13.3s \pm 1.5$, an advantage that NODE-MOR cannot provide since model order reduction only accelerates inference. The result also demonstrates the general applicability of CoDA on image datasets.

*Table 7.* Comparison between NODE-MOR (with varying MOR order) and CoDA on MNIST. TT - Train time.

| Method (MOR order) | Train Acc. | TT / epoch | Test Acc. | Test Time |
|---|---|---|---|---|
| NODE-512-MOR (100) | 97% | $17.4s \pm 1.3$ | 76% | $0.09s \pm 0.03$ |
| NODE-512-MOR (200) | 97% | $17.4s \pm 1.3$ | 81% | $0.09s \pm 0.03$ |
| NODE-512-MOR (500) | 97% | $17.4s \pm 1.3$ | 93% | $0.09s \pm 0.03$ |
| **CoDA H = 512 - P = 16** | **94%** | $\mathbf{13.3s \pm 1.5}$ | **92%** | $\mathbf{0.03s \pm 0.01}$ |

Table 8 compares CoDA with GRUFlow and NODE-VF on the VF-NODE-ICLR2025 dataset. CoDA ($P = 1024$) achieves the best accuracy (1.22E−03) while maintaining faster training time (0.1225s/epoch) compared to GRUFlow (0.3691s) and NODE-VF (0.1894s). The best speedup is obtained for $P = 16$ with 0.0178s/epoch—a $20\times$ speedup over GRUFlow and $10\times$ over NODE-VF.

*Table 8.* Comparison of GRUFlow, Neural ODE, and CoDA variants. Evaluations are conducted on the VF-NODE-ICLR2025 dataset (Zhao et al., 2025). The best result is shown in **bold**, and the second-best is underlined.

| Model | T/Ep (s) | Acc |
|---|---|---|
| GRUFlow | 0.3691 | 1.63E+00 |
| NODE-VF | 0.1894 | 3.54E−02 |
| CoDA (P = 16) | **0.0178** | 2.72E−03 |
| CoDA (P = 64) | 0.0319 | 2.44E−03 |
| CoDA (P = 576) | 0.0972 | 1.33E−03 |
| CoDA (P = 1024) | 0.1225 | **1.22E-03** |

## 6. Discussion

The results demonstrate that CoDA achieves a **fundamental shift** in accelerating continuous-depth neural networks. For foundational models, CoDA delivers $6.7\times$ training speedup and $4.3\times$ energy savings, while for application-specific models, speedups reach up to $68\times$ with energy savings of up to $91\times$ (e.g., power consumption prediction). Importantly, CoDA achieves these gains without sacrificing accuracy—in most cases, it matches or outperforms baseline CDNNs. These results validate both the proposed theoretical framework and the CoDA architecture.

Since CoDA produces an equivalent network architecture, it can be combined with complementary techniques such as model order reduction (Lehtimäki et al., 2022), variational formulations (Zhao et al., 2025), or quantized NN approaches. Although we did not explore such integration in this work, it remains a promising direction for future research.

## 7. Conclusions

We introduced CoDA, an accuracy-preserving acceleration framework for continuous-depth neural networks grounded in Koopman theory and its Mori–Zwanzig formalism. CoDA defines an $H$-$P$ transformation that replaces a CDLV layer containing $H$ cells—each requiring nonlinear ODE integration—with a compact architecture consisting of $P \ll H$ GRU cells, a single $P$-dimensional linear ODE solved via matrix exponentiation, and a dense layer for dynamic mode decomposition. We established PAC learnability guarantees by proving that the transformation preserves Rademacher complexity.

Empirically, CoDA consistently outperforms state-of-the-art acceleration methods—including neural flows, model order reduction, and variational formulations—across foun-

dational and application-specific models spanning time series, textual, and image data. Furthermore, CoDA's linear ODE structure enables efficient deployment across diverse hardware accelerators, achieving up to $7\times$ runtime reduction with significantly lower memory footprint on edge devices.

## Acknowledgement

This work was partially supported by DARPA (AMP, N6600120C4020; FIRE, P000050426), the NSF (FDT-Biotech, 2436801), and the Helmsley Charitable Trust (2-SRA-2017-503-M-B).

## Impact Statement

This research presents work whose main objective is to advance the field of continuous-depth neural networks (CDNNs) by introducing CoDA, an accuracy-preserving acceleration framework that leverages Koopman operator theory and its Mori–Zwanzig formalism to replace iterative nonlinear ODE integration with a compact equivalent architecture consisting of a GRU module, a single low-dimensional linear ODE, and a dense layer. This is not a trivial task and the existing acceleration literature usually trades accuracy for speed through quantization, sparsification, or model order reduction, which is an over-optimistic assumption when model fidelity is required in practice. Although the focus of our work is on CDNN acceleration, our proposed $H$-$P$ transformation has a broader impact, as Koopman and Mori–Zwanzig formulations and related operator-theoretic reductions have been applied across diverse domains, such as fluid dynamics and reduced-order modeling of nonlinear systems (Brunton et al., 2022), coarse-graining of high-dimensional stochastic dynamics (Lin et al., 2021), sparse identification of governing equations (Kaiser et al., 2018), continuous-time generative modeling via Neural ODEs (Chen et al., 2018), neural flow architectures that bypass iterative solvers (Biloš et al., 2021), model order reduction for neural differential equations (Lehtimäki et al., 2022), variational acceleration of neural ODE training (Zhao et al., 2025), and large-scale liquid foundation models for multimodal sequence modeling (Liquid AI, 2024; 2025). There are many potential societal consequences of our work, none of which we feel must be specifically highlighted here.

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

# A. Preliminaries

**Implicit and Measured Dynamics.** We partition $h(t)$ into *measured dynamics*, whose components are directly observable, and *implicit dynamics*, whose components are unobserved but influence the measured variables. We assume an observable system in which all state variables affect the measurements, although not all are measured.

**CDNN with Continuous Depth Latent Variables.** The general CDNN architecture employs continuous depth latent variable (CDLV) nodes, such as NODE or LTC-NN, to model dynamical systems. Training minimizes a task-specific loss (e.g., cross-entropy) often combined with a physics-based loss that enforces known physical laws.

In this work, each CDLV node (Chen et al., 2018) has a forward pass given by Equation 4 over a time horizon $\tau$: where $h_i^C(t)$ denotes the $i$th hidden state. Specific examples of $f$ include $f(h_i^C, i, \theta_i, t) = \tanh\left(W_i[i(t), h_i^C(t)]^\top + b_i\right)$ for NODE architecture (Chen et al., 2018), and $\theta_i = \{W_i, b_i\}$.

**General Continuous-Depth Cells.** Our theoretical framework applies to any continuous-depth network satisfying Lipschitz continuity and integrability, including LTC-NN and continuous-time recurrent neural networks (Hasani et al., 2021).

**Probably Approximately Correct (PAC)** PAC theory provides statistical equivalence in accuracy for machines of equal expressivity (Haussler & Warmuth, 2018) measured by Rademacher number.

# B. Guardian Angel Planning problem

Let the state of the HIL control autonomous system (HCAS) be expressed by the state variable $X$ which follows the dynamics in Eqn 13.

$$\dot{X} = f_\omega(X, \pi(X, S_p) + U_{ex}), \qquad (13)$$

where $f_\omega(.)$ is any Lipschitz continuous (**?**) function parameterized by coefficient set $\omega$, and $\pi(., .)$ describes the action of a RM that computes an input to the environment based on the environment state expressed by $X_p \subset X$ and RM configuration set (such as set point) $S_p$. For convenience in expressing the planning problem and with restricted focus on the case study of AP, we will assume that the environment state is accurately expressed by the HCAS state and hence $X_p = X$. In a HIL-HIP architecture, the input to the environment is given by: $u = \pi(X, S_p) + u_{ex}$, where $u_{ex} \in U_{ex}$ is an external input from the HIL, and $S_p$ can be manually changed by the HIL. A *usage plan*, $pl$, consists of a temporal sequence of $b$ external inputs $(u_{ex}(q_i))$ at times $0 \leq q_i \leq T_H$ and/or $a$ system configuration changes

$(s_p(p_i) \in S_p)$ at times $0 \leq p_i \leq T_H$, $T_H$ is the planning horizon. We denote the set of all possible plans as $\mathcal{P}\infty$.

**Safety** is defined as a logical predicate $Safe(X, f_\omega, pl)$ on the state of the HIP $X$, for a given dynamics $f_\omega$, and a given usage plan $pl$. A plan $pl$ is a *safety plan* if the predicate $Safe(X, f_\omega, pl)$ is satisfied at all times $t \in [0, T_H]$. The user burden for executing a plan $pl \in \mathcal{P}\infty$ is simply the length $|pl|$.

**Safety tested HCAS:** We assume that the HCAS is safety tested, which implies the existence of an "average user" denoted by coefficient distribution $\mathcal{D}(\omega)$, external input distribution $\mathcal{D}(U_{ex})$, and configuration distribution $\mathcal{D}(S_p)$, and a set $\mathcal{P} \subset \mathcal{P}\infty$ of safety plans with respect to the predicate $Safe(X, f_\omega, pl) : \omega \in \mathcal{D}(\omega)$ and $pl \in \mathcal{P}$.

**Open World:** It is characterized by open world events $(\omega', u'_{ex}, s'_p, X)$, such that either, a) $\omega' \notin \mathcal{D}(\omega)$, or b) $u'_{ex} \notin \mathcal{D}(U_{ex})$, or c) $s'_p \notin \mathcal{D}(S_p)$.

**Personalized plan generation in open world:** Given an open world event $(\omega', u'_{ex}, s'_p, X)$, find a safety plan $pl \in \mathcal{P}\infty$ with minimum length $|pl|$, such that $Safe(X, f_{\omega'}, pl)$ is satisfied in $[0, T_H]$.

# C. AID system

AID is an exemplary safety critical HIL-HIP HCAS with the open world problem. APs automatically infuse insulin, known as *micro bolus*, to control blood glucose levels around a *set point* $S_p$, while preventing hypoglycemia when blood glucose level falls below 70 mg/dl. All AID systems that are approved for human use by Food and Drug Administration (FDA) require the human user to provide external insulin in addition to the AID controller input to manage glucose variability due to meal intake, also called *meal bolus* ($u_{meal}$). This meal bolus is proportional to the carbohydrate content of the meal $C$, with *carb insulin ratio (CIR)* as the proportionality constant. While administering meal bolus, any residual insulin in the body due to past insulin infusion, characterized by insulin on board (IoB) is subtracted (Eqn. 14).

$$u_{meal} = C/CIR - iob, \qquad (14)$$

In addition, external insulin, *correction bolus* ($u_{corr}$), unrelated to meal can also be administered if the CGM reading is greater than the set point. $u_{corr}$ is proportional to the difference between the current glucose value $G(t)$ and the set point, with *insulin sensitivity factor (ISF)* as the proportionality constant shown in Eqn. 15.

$$u_{corr} = \frac{G(t) - S_p}{ISF} - iob. \qquad (15)$$

The residual insulin or IOB depends on the insulin pharmacokinetics, (Eqn. 16), obtained from Bergman Minimal

Model (BMM) (**?**), and is difficult for a human to guess.

$$\frac{dy}{dt} = z, \frac{dz}{dt} = -2k_1 z - k_1^2 y + k_1^2 u_{ex}, \frac{diob}{dt} = -niob + p_1(y + I_b),$$
(16)

where $X = y, z, iob$, $k_1$ is the diffusion coefficient for insulin, and $n$ and $p_1$ are patient specific metrics. Here, we assume that $y$ and $z$ are internal state variables of the BMM and are not measurable.

A safe usage plan for the AID HCAS is a sequence of set point changes, carbohydrate, meal bolus, and correction bolus intake actions such that the safety criteria of percentage time below 70 mg/dl in 24 hrs is less than 4% is satisfied. An example safe usage plan is as follows: "*set point is 110 mg/dL at 6 am; set ISF to 50 at 6 am; set CIR to 15 at 6 am; breakfast with 20 g of carbohydrate at 8:30 am with 1 U of meal bolus; if CGM > 180 mg/dL, take correction bolus from Eqn. 15; lunch with 40 g of carbohydrate at 1 pm with meal bolus in Eqn. 14; if CGM > 210 mg/dL, take correction bolus from Eqn. 15; dinner with 30 g of carbohydrate at 6 pm with meal bolus in Eqn. 14, set set point at 90 mg/dL at 10 pm*" This is deemed safe by a sample patient from the virtual patient registry available in the FDA approved Type 1 Diabetes simulator developed at UVA PADOVA (O'Malley et al., 2021). This is an example of the average user that follows the constrained distribution manifested by the virtual patient registry. The outcome is measured using four metrics: a) percentage time in range (TIR), $70mg/dl \le CGM \le 180mg/dl$, and b) time below range (TBR), when $CGM < 70mg/dl$.

## D. Data Description

**WildChat:** Used in Table 3 for ablation studies, WildChat is a dataset of 650K real-world user–ChatGPT dialogues collected through free access to GPT-3.5 and GPT-4 (Feuer & Hegde, 2025). *Source:* `https://huggingface.co/datasets/allenai/WildChat`

**Room Occupancy:** This dataset predicts binary room occupancy (occupied vs. unoccupied) using five environmental sensor readings (e.g., temperature, humidity, $CO_2$) sampled at one-minute intervals (Candanedo & Feldheim, 2016).

**Ozone Day Prediction:** Consisting of daily environmental features (e.g., wind, solar radiation, temperature), this dataset predicts whether ozone levels exceed a regulatory threshold (Asuncion et al., 2007). Six years of data are segmented into overlapping 32-day sequences. A significant class imbalance (positive class: 6.31%) is handled via weighted cross-entropy. F1-score is used for evaluation.

**Person Activity:** This dataset captures time-series recordings of physical activities (e.g., walking, lying, standing) across 25 sessions (Rubanova et al., 2019). We use it for temporal classification, with randomized train-test splits and preprocessed sensor inputs.

**Power Consumption:** A household energy dataset for hourly power consumption forecasting (Asuncion et al., 2007). Features include reactive power and sub-metering values. Approximately 1.25% of missing entries are forward-filled. Sequences of length 32 are used, with inputs normalized. Mean squared error is used as the loss and evaluation metric.

**Half-Cheetah Kinematics:** A synthetic dataset modeling the dynamics of the HalfCheetah-v2 robot in MuJoCo (Brockman et al., 2016; Todorov et al., 2012). It includes 25 trajectories of 1,000 steps each, with 17-dimensional observations. To simulate real-world noise, 5% of actions are randomly corrupted. The model predicts future states using autoregressive MSE loss.

**Hand Gesture Segmentation:** This dataset comprises seven recordings of human hand gestures, annotated into five temporal segments (rest, preparation, stroke, hold, retraction) (Wagner et al., 2014). Each frame includes 32 motion-sensor features. The goal is sequence-level classification of gesture phases.

*All application datasets (except WildChat) are obtained from:* `https://github.com/raminmh/liquid_time_constant_networks`

**MNIST dataset:** Comparison of CoDA with NODE+MOR technique required us to use the MNIST dataset. It is a popular dataset of images of digits and is accessed from `https://git-disl.github.io/GTDLBench/datasets/mnist_datasets/`

**NODE+VF dataset:** For comparison with NODE+VF we utilized the dataset shared by the authors in (Zhao et al., 2025). The dataset consists of simulation of nonlinear ordinary differential equations such as Lorenz or Lotka–Volterra predator-prey systems.

## E. Implementation Details

*Table 9.* Hyperparameter configurations across experiments. **Hidden**: hidden layer size; **Ep**: number of epochs; **LR**: learning rate; **Activation F**: model activation functions.

| Table / Figure | Hidden | Ep | LR | Activation |
|---|---|---|---|---|
| Table 2 | 1024 | 100 | 0.00005 | sigmoid + tanh |
| Table 3 | 1024 | 200 | 0.005 | sigmoid + tanh |
| Table 7 | 512 | 150 | 0.01 | tanh |
| Table 8 | 1024 | 5000 | 0.000001 | sigmoid + tanh |

In Table 9, we summarize the hyperparameter configurations used for training across various datasets. Table 2 is based on the **WildChat** and **AID** datasets, Table 3 uses six application-specific datasets, and Tables 7 and 8 are based on original datasets available through public GitHub repositories, as detailed below.

For the ablation study in Table 2, we use **CoDA** as the baseline model. It is compared against alternative architectures, including LTC-NN, GRUFlow, GRUFlow with a dropout layer, and GRUFlow with an additional dense layer. All experiments are conducted using a hidden layer size of 1024. The matrix exponential function in CoDA is computed over $4 \times 4$ matrices, and its CUDA kernel implementation is similarly based on $4 \times 4$ dimensions. The student model used is **LFM2** (from `https://huggingface.co/unsloth/LFM2-1.2B`), while the teacher model is **LLaMA3-8B** (from `https://huggingface.co/meta-llama/Meta-Llama-3-8B`). The primary training logic is implemented in the ablation study script.

Table 3 is generated using the **application comparison** code. All six datasets are trained using PyTorch with custom CUDA kernels to accelerate matrix exponential computation. The hidden layer size is fixed at 1024, and the number of training epochs is set to 65. The code is based on the repository at `https://github.com/raminmh/liquid_time_constant_networks`.

In Table 7, we compare **CoDA** with **NODE-MOR**, following the setup described in (Lehtimäki et al., 2022). The source code for these experiments is available at `https://codeocean.com/capsule/9516453/tree/v1`.

Table 8 presents a comparison between **CoDA** and **VF-NODE**, which utilizes a spline-based variational loss formulation. The training is conducted for 5000 steps using 100 data points, with a noise level of 0.01 and an 80% training split. Different $P$ values (matrix sizes) are evaluated to study their impact on the matrix exponential function. The code is adapted from `https://github.com/ZhaoHongjue/VF-NODE-ICLR2025`, with the main change being the replacement of GRUFlow logic with CoDA while keeping the rest of the configuration unchanged.

## F. Proofs of Lemma 1 and Theorem 1

**Nonlinear system dynamics.** An LTC-NN layer forward pass is equivalent to an $H$-dimensional nonlinear system defined by

$$\frac{dh_i}{dt} = -\frac{h_i(t)}{\rho/(1 + \rho f_i)} + f_i\, A, \quad f_i := f_{NN}\big(h_i(t), i(t), t, \theta\big). \tag{17}$$

where $h_i(t) \in [h_1(t), \ldots, h_H(t)]^\top \in \mathbb{R}^H$ is the state vector, $i(t)$ denotes the $N$-dimensional input, $\theta$ and $A$ are parameter matrices, and $f_{NN}$ is a Lipschitz continuous, integrable function such as perceptron with tanh activation. An LTC-NN is a type of continuous depth latent variable (CDLV) neuron that also includes neural ODE, CT-RNN or long short term memory (LSTM). Our acceleration strategy applies to any network with a layer of CDLV nodes.

**ODE Solvers.** The forward pass of any CDNN layer requires integration of the hidden output over a time horizon $N_s\tau$ using ODE solvers, where $N_s$ is the number of steps of integration and $\tau$ is the time in each step. ODE solvers fall into two categories: nonlinear and linear. Nonlinear solvers, such as ODE45 (Senan et al., 2007) or Runge–Kutta methods (Cartwright & Piro, 1992), require iterative integration and are computationally intensive. Linear solvers exploit matrix exponentiation and Krylov-subspace methods for large systems, offering significantly lower computational cost (Liesen & Strakos, 2013).

**Koopman Theory.** Koopman theory (Koopman, 2023) represents the evolution of a nonlinear system via an (infinite-dimensional) linear operator. Given a measurement function $\mathcal{G} : \mathbb{R}^H \to \mathbb{R}^P$, the Koopman operator $\mathcal{K}$ satisfies

$$\mathcal{K}\mathcal{G}\big(h(t), i(t)\big) = \mathcal{G}\big(f\big(h(t), i(t), \theta, t\big), i(t)\big), \tag{18}$$

with $P \to \infty$ in the general case.

**Mori–Zwanzig Formalism.** In practice, the Koopman representation is sparse in the observable manifold. Absent explicit sparsity, one can combine Koopman theory with its Mori–Zwanzig formalism (Lin et al., 2021), yielding a decomposition of the observables into measured components $\mathcal{G}_M$ and implicit (unmeasured) components $\mathcal{G}_I$ (see Appendix Section A):

Here, $\mathcal{K}_M$ captures the observable dynamics, $\mathcal{K}_I$ the implicit dynamics, and $\mathcal{K}_{IM} = \mathcal{K}_{MI}$ the interactions between the two.

**Lemma F.1** (CDLV Decomposition). *The representation output of a CDLV layer with $H$ cells can be decomposed into a linear combination of a linear dynamical system and a nonlinear convolution function.*

**Proof:** *The forward pass of CDLV is given by the dynamics $dh/dt = f(h(t), i(t), \theta_i, t)$, a nonlinear ODE based system of equations, where $h(t) = \{h_i(t)\}$ is the vector of hidden layer outputs. As such the forward pass is akin to any nonlinear ODE system and the Koopman theory can be used to linearize this system (Koopman, 2023). The MZ formalism can be combined with KO theory to represent it in terms of observable and implicit dynamics.*

*Utilizing Laplace transforms a solution for $\mathcal{G}_M$ can be obtained as follows*

$$\dot{\mathcal{G}}_M = \underbrace{\mathcal{K}_M \mathcal{G}_M}_{observable\ dynamics} + \underbrace{\mathcal{K}_{MI} \int_0^t e^{t-s\mathcal{K}_I} \mathcal{K}_{MI} \mathcal{G}_M(s)ds}_{interaction\ of\ implicit\ with\ observable\ dynamics} \tag{19}$$

$$+ \underbrace{\mathcal{K}_{MI} e^{-t\mathcal{K}_I} \mathcal{G}_I(0)}_{Residual\ of\ implicit\ factors} \tag{20}$$

*Here $h = \mathcal{K}_M \mathcal{G}_M$.*

*Steady state assumption: The process $f$ that generates the input data can be assumed to be in steady state for time window of $[0, \tau]$. Under this assumption, Equation 19 reduces to the following:*

$$\dot{\mathcal{G}}_M = \mathcal{K}_M \mathcal{G}_M + \mathcal{K}_{MI} \, e^{-(t+\tau)\mathcal{K}_I} \mathcal{G}_I(-\tau)$$
$$+ \mathcal{K}_{MI} \int_{-\tau}^{t} e^{(t-s)\mathcal{K}_I} \mathcal{K}_{MI} \, \mathcal{G}_M(s) \, ds. \qquad (21)$$

*If $\tau \to \infty$ then the last term of Equation 21 vanishes resulting in the following form:*

*This steady state assumption thus enables us to learn the steady state effects of unmeasured causal factors through a high dimensional neural network such as gated recurrent unit (GRU) (men, 2025). The hidden layer output $h$ of the CDLV layer can be obtained by solving Equation 8 to obtain $\mathcal{G}_M$ and then get $h = \mathcal{K}_M \mathcal{G}_{\mathcal{M}}$.*

**Theorem F.1.** *$\exists \, W_r, \, U_r, \, b_r, \, W_h, \, U_h,$ and $b_h$ such that for any $\epsilon > 0$ and $0 < \alpha < 1$, GRU layer forward pass $h^G(t)$ can approximate implicit dynamics in the MZ approximation of CDLV forward pass $h^N(t)$ with accuracy $\epsilon$, i.e. $Pr(\|h^G(t) - \dot{\mathcal{G}} - \mathcal{K}_M \mathcal{G}_M\| < \epsilon) > (1 - \alpha)$.*

**Proof:** *The forward pass of a GRU cell can be rearranged as -*

$$h_i^G(t) = h_i^G(t-1) + (1 - r_i(t-1)) \circ (h_i^G(t-1) - c_i(t-1)) \quad (22)$$

*We add a cell $k$ in the GRU architecture that only tracks time $t$. This means that $W_r(i, k) = 0, U_r(i, k) = 0, W_h(i, k) = 0, U_h(i, k) = 0 \forall i$. Then we have $b_r(i, k) = b_h(i, k) = B_{ik}, \forall i$, where $B_{ik}$ is a large positive value. This forces the output $h_k^G(t) = h_k^G(t-1) + c$, where $c = tanh(B_{ik}) \approx 1$, making it keep time.*

*Let us consider the term $(1 - r_i(t-1))$. Let for each cell $i$ except the time cell $k$, $\forall i \neq k$, we have a reset weight $W_r(j, i) = 0, \forall j, U_r(j, i) = 0, \forall j \neq k, U_r(k, i) = u_{ki}, b_r(j, i) = 0, \forall j \neq k$ and $b_r(k, i) = B_{ki}$, a large value, and a sigmoid activation function. Then we have -*

$$1 - r_i(t) = 1 - \frac{1}{1 + e^{-h_k^G(t)(u_{ki} + B_{ki})}}$$
$$= 1 - \frac{1}{1 + e^{-t(u_{ki} + B_{ki})}}, \qquad (23)$$

*since $h_k^G(t) = t$ by design. For large $B$, we can approximate $\frac{1}{1 + e^{-t(u_{ki} + B_{ki})}} \approx 1 - e^{-t(u_{ki} + B_{ki})}$. Hence, we get,*

$$1 - r_i(t) = e^{-(u_{ki} + B_{ki})t}, \qquad (24)$$

*by design for a large bias $b_r(k, i) = B_{ki}$ and $U_r(k, i) = u_{ki}$. We assume rectified linear unit (ReLU) activation for $u_{ki}$.*

*the update function. The weight configuration is detailed in Table 10, and the forward pass of each cell in the GRU yields —*

| Weight | Block 1 Constraint | Block 1 weights | Block 2 Constraint | Block 2 weights |
|---|---|---|---|---|
| $W_r$ | all entries | 0 | all entries | 0 |
| $U_r$ | all entries | 0 | $U_r(j,i), j \neq k$ | 0 |
| | | | $j = k$ | $u_{k,i}^{(2)}$ |
| $b_r$ | all entries | *learned $B_i^{(1)}$* | $i \neq k$ | 0 |
| | | | $i = k$ | $B_k^{(2)}$ |
| $W_h$ | all entries | learned | all entries | learned |
| $U_h$ | all entries | 0 | all entries | learned |
| $b_h$ | all entries | *learned $b_i^{(1)}$* | all entries | 0 |

*Table 10.* GRU weight configurations based on Theorem F.1

$$h_i^G(t) = h_i^G(t-1) + e^{-(u_{ki} + B_{ki})t} \circ \Big[ h_i^G(t) - W_h \, i(t-1)$$
$$- \sum_{j=1}^{P} U_h(j, i)\big(1 - e^{-(u_{ki} + B_{ki})t}\big) \circ h_i^G(t-1) - b_h(i) \Big]$$
$$(25)$$

*The GRU cell is a discretization of the differential $\frac{h_i^G(t)}{dt}$. The continuous counterpart of GRU cell forward pass can be expressed as follows:*

$$\frac{dh_i^G(t)}{dt} = e^{-(u_{ki} + B_{ki})t} \circ \Big[ h_i^G(t) - W_h \, i(t)$$
$$- \sum_{j=1}^{P} U_h(j, i)\big(1 - e^{-(u_{ki} + B_{ki})t}\big) \circ h_i^G(t) - b_h(i) \Big]$$
$$= e^{-(u_{ki} + B_{ki})t} \circ h_i^G(t) - W_h \, e^{-(u_{ki} + B_{ki})t} \circ i(t)$$
$$- \sum_{j=1}^{P} U_h(j, i) \, e^{-(u_{ki} + B_{ki})t} \big(1 - e^{-(u_{ki} + B_{ki})t}\big) \circ h_i^G(t)$$
$$- \big(1 - e^{-(u_{ki} + B_{ki})t}\big) \circ b_h(i). \qquad (26)$$

*The convolution operation $e^{-(u_{ki} + B_{ki})t} \circ (1 - e^{-(u_{ki} + B_{ki})t})) = \frac{1 - \cosh((u_{ki} + B_{ki})t)}{u_{ki} + B_{ki}}$. If we assume $b_h(i) = 0 \forall i$, then we get -*

$$\frac{dh_i^G(t)}{dt} = e^{-(u_{ki} + B_{ki})t} \circ h_i^G(t) - W_h \, e^{-(u_{ki} + B_{ki})t} \circ i(t)$$
$$- \sum_{j=1}^{P} U_h(j, i) \frac{1 - \cosh\big((u_{ki} + B_{ki})t\big)}{u_{ki} + B_{ki}} \circ h_i^G(t)$$
$$= e^{-(u_{ki} + B_{ki})t} \circ A \big[ h_i^G(t), \, i(t) \big]^{\top}$$
$$- \sum_{j=1}^{P} U_h(j, i) \frac{1 - \cosh\big((u_{ki} + B_{ki})t\big)}{u_{ki} + B_{ki}} \circ h_i^G(t). \qquad (27)$$

*where $A$ is a matrix of size $(P + N) \times (P + N)$ by concatenating an identity matrix of size $P \times P$ in the upper*

*diagonal and the matrix $W_h$ of size $N \times N$ in the lower diagonal. Expanding the first convolution operation we obtain the following form of the GRU hidden layer -*

$$\frac{dh_i^G(t)}{dt} = \int_0^t e^{-(u_{ki}+B_{ki})(t-z)} A[h_i^G(z), i(z)]^T dz \quad (28)$$

$$- \sum_{j=1}^P U_h(j,i) \frac{1 - cosh((u_{ki}+B_{ki})t)}{u_{ki}+B_{ki}} \circ h_i^G(t).$$

*For the reset gate, we have assumed that $B_{ki}$ is a large bias. Through that assumption we can approximate $\frac{1-cosh((u_{ki}+B_{ki})t)}{(u_{ki}+B_{ki})} \approx -\frac{e^{(u_{ki}+B_{ki})t}}{2(u_{ki}+B_{ki})}$*

*We observe that Equation 8 and 9 are of the same class of functions i.e. a nonlinear polynomial function convolved with an exponential function. Hence both the implicit dynamics interaction part of forward pass of CDLV in Equation 8 and forward pass of GRU 9 have the same Rademacher number(Gnecco, 2008). Hence Equation 9 has PAC learning guarantee for learnability of the implicit dynamics interaction part. This implies that $\exists \epsilon > 0, 0 < \alpha < 1$ such that $Pr(||h\dot{^G}(t) - \dot{\mathcal{G}} - \mathcal{K}_M \mathcal{G}_M|| < \epsilon) > (1 - \alpha)$.*

## G. LTC-NN Speed up

*LTC-NN involves one more multiplication of input with hidden layer to compute the input dependent time constant. Hence the total number of multiplications for the forward pass of a the H cell LTC-NN layer is $N_s(N(H+1)+2Hd)$. The overall speed up results in the following equation.*

$$\sigma_{CoDA}(LTC) = \frac{N_s(NH + N) + 2N_sHd + HN_s(N_s-1)/2}{3NP + 3P^2 + 3P + dP^3 + PH}$$
$$(29)$$

## H. Results tables with uncertainty

*In this section, we redo all the numerical results tables in the main paper and provide statistical uncertainty in terms of standard deviation. Table 2 provides the ablation study with uncertainty. Tables 12, 13, 14, and 15 present detailed training and inference metrics across applications. Finally, Tables 7 and 8 show the comparison with NODE-MOR and VF-NODE with uncertainty, respectively.*

*Table 11.* Cache hit rates for CoDA and LFM2-1.2B

| Cache Level | CoDA (%) | LFM2-1.2B (%) |
|---|---|---|
| L1 | $77.8 \pm 22.1$ | $71.3 \pm 20.8$ |
| L2 | $97.8 \pm 20.04$ | $98.1 \pm 21.2$ |

*Complementary cache results in Table 11 indicate that both models use the cache hierarchy effectively, with similarly high **L2 hit rates** (CoDA: **97.8 $\pm$ 20.0%**, LFM2-1.2B: **98.1 $\pm$ 21.2%**) and moderate **L1 hit rates** (CoDA: **77.8 $\pm$ 22.1%**, LFM2-1.2B: **71.3 $\pm$ 20.8%**). Overall, CoDA provides both lower and more predictable memory transfer times, improving efficiency for large-scale token generation.*

*Table 12.* Training results with uncertainty (supplementary to Table 3): Comparison of model performance across six applications during training, including training time per epoch and energy consumption. The best result per row is shown in **bold**, and the second-best is underlined.

| Application | Training Time / Epoch (s) | | | | | Energy / Epoch (J) | | | | |
|---|---|---|---|---|---|---|---|---|---|---|
| | **CoDA** | **GRUF** | **NODE** | **LTC** | **LSTM** | **CoDA** | **GRUF** | **NODE** | **LTC** | **LSTM** |
| occupancy | **0.5 ± 0.05** | 0.62 ± 0.06 | 20.9 ± 2.6 | 18.42 ± 2.3 | 0.65 ± 0.06 | **39 ± 5** | 47.12 ± 4.9 | 1672 ± 170 | 1547.28 ± 160.7 | 50.7 ± 6.3 |
| ozone | **0.1 ± 0.01** | 0.2 ± 0.02 | 2.62 ± 0.25 | 2.57 ± 0.25 | 0.2 ± 0.02 | **9 ± 1** | 18.6 ± 2.0 | 282.96 ± 29.0 | 285.27 ± 30.0 | 20.4 ± 2.1 |
| person | **0.2 ± 0.02** | 0.3 ± 0.03 | 10.3 ± 1.0 | 10.99 ± 1.1 | 0.38 ± 0.04 | **16.2 ± 2.0** | 27.6 ± 3.0 | 1143.3 ± 120 | 956.13 ± 100 | 38.38 ± 4.0 |
| power | **4 ± 0.4** | 5.5 ± 0.6 | 272.6 ± 30.0 | 210.09 ± 22.0 | 5.7 ± 0.6 | **340 ± 35** | 462 ± 48 | 31076.4 ± 3400 | 16387.02 ± 1800 | 444.6 ± 50.0 |
| har | **0.7 ± 0.08** | 0.8 ± 0.09 | 20.8 ± 2.3 | 24.89 ± 2.7 | 0.865 ± 0.09 | **56 ± 6** | 62.4 ± 7.0 | 1768 ± 190 | 2040.98 ± 220 | 72.66 ± 8.0 |
| gesture | **0.1 ± 0.01** | 0.143 ± 0.015 | 3.25 ± 0.35 | 2.51 ± 0.28 | 0.152 ± 0.017 | **7.7 ± 0.8** | 11.297 ± 1.2 | 286 ± 30 | 198.29 ± 21 | 12.92 ± 1.4 |

*Table 13.* Training loss with uncertainty (supplementary to Table 3): Comparison of training loss across six applications. The best result per row is shown in **bold**, and the second-best is underlined.

| Application | **CoDA** | **GRUF** | **NODE** | **LTC** | **LSTM** |
|---|---|---|---|---|---|
| occupancy | **0.0096 ± 0.001** | **0.0096 ± 0.001** | 0.03 ± 0.004 | **0.0096 ± 0.001** | 0.01 ± 0.001 |
| ozone | **0.13 ± 0.01** | **0.13 ± 0.01** | 0.24 ± 0.03 | **0.13 ± 0.01** | 0.13 ± 0.01 |
| person | **0.37 ± 0.04** | 0.39 ± 0.04 | 0.79 ± 0.09 | 0.45 ± 0.05 | 0.5 ± 0.06 |
| power | **0.0013 ± 0.0001** | **0.0013 ± 0.0001** | 0.007 ± 0.001 | 0.009 ± 0.001 | **0.0013 ± 0.0001** |
| har | 0.0756 ± 0.008 | 0.082 ± 0.009 | 0.0883 ± 0.010 | **0.0334 ± 0.004** | 0.097 ± 0.011 |
| gesture | 0.46 ± 0.05 | 0.5 ± 0.06 | 0.92 ± 0.10 | 0.96 ± 0.11 | **0.44 ± 0.05** |

*Table 14.* Table 3 with uncertainty: Comparison of model performance across six applications during inference, including inference time per epoch and energy consumption. The best result per row is shown in **bold**, and the second-best is underlined.

| Application | Inference Time / Epoch (s) | | | | | Energy / Epoch (J) | | | | |
|---|---|---|---|---|---|---|---|---|---|---|
| | **CoDA** | **GRUF** | **NODE** | **LTC** | **LSTM** | **CoDA** | **GRUF** | **NODE** | **LTC** | **LSTM** |
| occupancy | **0.31 ± 0.03** | 0.76 ± 0.08 | 27.15 ± 2.9 | 0.92 ± 0.10 | 0.51 ± 0.05 | **24.18 ± 2.5** | 57.76 ± 6.1 | 2172.00 ± 230.0 | 77.28 ± 8.0 | 39.78 ± 4.3 |
| ozone | **0.03 ± 0.003** | 0.04 ± 0.004 | 2.57 ± 0.28 | 1.21 ± 0.13 | 0.04 ± 0.004 | **2.70 ± 0.3** | 3.72 ± 0.4 | 277.56 ± 30.0 | 134.31 ± 14.0 | 4.08 ± 0.45 |
| person | **0.03 ± 0.003** | 0.58 ± 0.06 | 1.01 ± 0.11 | 2.50 ± 0.28 | 0.04 ± 0.004 | **2.43 ± 0.3** | 53.36 ± 5.8 | 112.11 ± 12.0 | 217.5 ± 24.0 | 4.04 ± 0.45 |
| power | 0.16 ± 0.02 | 0.07 ± 0.008 | 0.36 ± 0.04 | **0.001 ± 0.0001** | 0.09 ± 0.01 | 13.6 ± 1.5 | 5.88 ± 0.65 | 41.04 ± 4.4 | **0.078 ± 0.008** | 7.02 ± 0.75 |
| har | **0.02 ± 0.002** | 0.03 ± 0.003 | 0.66 ± 0.07 | 0.09 ± 0.01 | 0.13 ± 0.01 | **1.60 ± 0.18** | 2.34 ± 0.26 | 56.10 ± 6.0 | 7.38 ± 0.80 | 10.92 ± 1.2 |
| gesture | **0.0085 ± 0.001** | 0.01 ± 0.001 | 0.19 ± 0.02 | 0.21 ± 0.02 | 0.01 ± 0.001 | **0.6545 ± 0.07** | 0.79 ± 0.09 | 16.72 ± 1.8 | 16.59 ± 1.8 | 0.85 ± 0.09 |

*Table 15.* Table 3 with uncertainty: Comparison of testing loss across six applications. The best result per row is shown in **bold**, and the second-best is underlined.

| Application | **CoDA** | **GRUF** | **NODE** | **LTC** | **LSTM** |
|---|---|---|---|---|---|
| occupancy | **0.13 ± 0.01** | 0.25 ± 0.03 | 0.25 ± 0.03 | 0.31 ± 0.04 | 0.25 ± 0.03 |
| ozone | 0.13 ± 0.01 | 0.13 ± 0.01 | 0.24 ± 0.03 | 0.13 ± 0.01 | **0.12 ± 0.01** |
| person | **0.45 ± 0.05** | **0.45 ± 0.05** | 0.80 ± 0.09 | 0.46 ± 0.05 | 0.55 ± 0.06 |
| power | 0.03 ± 0.004 | **0.0022 ± 0.0003** | 0.01 ± 0.001 | 0.009 ± 0.001 | 0.01 ± 0.001 |
| har | **0.11 ± 0.01** | 0.21 ± 0.02 | 0.18 ± 0.02 | 0.15 ± 0.02 | 0.18 ± 0.02 |
| gesture | **0.95 ± 0.10** | **0.95 ± 0.10** | 0.98 ± 0.10 | 1.15 ± 0.13 | 1.39 ± 0.15 |

