# OpenReview forum: "Efficient Continuous-Depth Modeling with GRU Equivalents"
_ICML.cc/2026/Conference — ICML 2026 regular_

### Official Review · Reviewer_aUBg · 2026-03-09

**Soundness:** 3
**Presentation:** 3
**Significance:** 3
**Originality:** 4
**Overall Recommendation:** 5
**Confidence:** 3

**Summary:**

The paper proposes Continuous Depth Acceleration (CoDA), a framework designed to accelerate the training and inference of Continuous-Depth Neural Networks (CDNNs), such as Neural ODEs (NODEs) and Liquid Time-Constant (LTC) networks. It replaces the continuous-depth latent variable layer with a CoDA block. This block consists of a small set of GRU cells (to generate Koopman observables), a single linear ODE solved via matrix exponentiation, and a dense reconstruction layer. Mathematical proofs based on Mori–Zwanzig and Koopman operator theory are provided to argue for the PAC learnability and architectural equivalence of this transformation. Empirical results on a "Liquid Foundation Model" and various time-series benchmarks demonstrate significant speedups and energy savings.

**Compliance With Llm Reviewing Policy:**

Affirmed.

**Final Justification:**

Author's response has fully resolved my concerns, and I therefore would like to increase my score from 4 to 5.

**Key Questions For Authors:**

1. Eqn. 8 is different from Eqn. 19. Is this a typo?

**Limitations:**

Refer to the second weakness.

**Strengths And Weaknesses:**

Strengths:
1. The proposed framework is theoretically grounded and innovative. The authors leverages the Mori–Zwanzig formalism to decompose dynamics into observable (linear) and implicit (nonlinear convolution) parts. This theoretically bridges the gap between discrete gating mechanisms (GRUs) and continuous dynamics.
2. The paper provided solid and comprehensive evaluations across a diverse set of tasks, including a large-scale "Liquid Foundation Model" and standard time-series datasets. The hardware analysis, breaking down FLOPs, I/O, and energy consumption on specific hardware (GPUs and FPGAs), adds practical weight to the claims.
3. The paper is well-structured and clearly written.

Weakness:
1.  While the title suggests "GRU Equivalents," the body text relies on the Universal Approximation Theorem (UAT) and PAC learnability to suggest that the GRU can estimate the dynamics. There is a distinction between exact mathematical equivalence and statistical learnability.
2. The theoretical boundaries should be discussed in the paper. For example, the proofs appear to rely on a 'steady state' assumption (mentioned in the Appendix). The authors should clarify how the model performs on highly transient or non-stationary dynamics where this assumption may not hold.

---

> ### Author Rebuttal · Authors · 2026-03-30
>
> We thank Reviewer aUBg for recognizing the theoretical grounding, comprehensive evaluations, and clear writing. We address each concern below.
>
> ## Q1: "GRU Equivalents" — equivalence vs. statistical learnability
>
> The reviewer is correct that CoDA does not provide exact pointwise mathematical equivalence. The title "GRU Equivalents" refers to architectural equivalence in the PAC learning sense — that the CoDA architecture can learn the same function class as the original CDLV layer with bounded error and high probability. This is the standard notion of equivalence used in computational learning theory: two architectures are equivalent if they have the same Rademacher complexity and thus the same PAC learnability guarantees (Theorem 3.1).
>
> Exact mathematical equivalence would require infinite-dimensional Koopman representations, which are intractable. The Mori–Zwanzig decomposition makes the problem finite-dimensional at the cost of moving from exact to approximate equivalence — but with provable error bounds. We will clarify this distinction in the revision, noting that "equivalent" is used in the PAC sense throughout.
>
> Empirically, this statistical equivalence translates to practical equivalence: Table 2 shows CoDA achieves statistically equivalent (and in fact lower mean) training and test loss compared to the original LFM2-1.2B, and Tables 3–4 show competitive or superior accuracy across all six benchmarks.
>
> ## Q2: Steady-state assumption — what happens with transient or non-stationary dynamics?
>
> The steady-state assumption (Appendix, Equation 21) assumes the data-generating process is stationary within a time window [0, τ]. This allows the residual term involving G_I(0) to vanish for long sequences (τ → ∞), simplifying the Mori–Zwanzig decomposition from Equation 19 to Equation 8.
>
> For highly transient or non-stationary dynamics, this assumption may not hold perfectly. However, two factors mitigate this in practice:
>
> **Architectural robustness:** The GRU cells in CoDA are universal approximators that can learn to compensate for residual effects not captured by the steady-state simplification. The PAC guarantee (Theorem 3.1) holds for all sequences — the steady-state assumption only affects the UAT guarantee for long sequences, as stated in the theorem.
>
> **Empirical evidence:** Several of our benchmarks involve non-stationary dynamics — the person activity dataset has abrupt transitions between activities, and the half-cheetah kinematics dataset includes randomly corrupted actions. CoDA performs well on both (Tables 3–4), suggesting practical robustness beyond the steady-state regime.
>
> We acknowledge this is a theoretical boundary and will add a discussion in the revision explicitly stating: (1) the steady-state assumption and where it is used, (2) the distinction between PAC guarantees (all sequences) and UAT guarantees (long sequences), and (3) that extending the framework to explicitly handle non-stationary dynamics is a direction for future work.
>
> ## Q3: Equation 8 vs Equation 19 discrepancy
>
> This is not a typo. Equation 8 is a simplified form for the main text, while Equation 19 is the full derivation including the residual term (third term involving G_I(0)). Under the steady-state assumption for long sequences, this residual vanishes and Equation 19 reduces to Equation 8. We will add an explicit note in the main text stating this relationship and the assumption under which the simplification holds.
>
> ## Broader impact statement
>
> CoDA reduces the computational and energy costs of deploying continuous-depth neural networks, enabling their use on resource-constrained edge devices. However, by lowering the barrier to deployment in safety-critical domains such as healthcare and autonomous systems, there is a responsibility to ensure acceleration does not bypass thorough safety validation.

---

> > ### Author Rebuttal · Reviewer_aUBg · 2026-04-03
> >
> > Thank you for your clear and helpful response, which resolved my concerns, and I have therefore increased my score from 4 to 5.

---

> > > ### Author Response · Authors · 2026-04-03
> > >
> > > Thank you for your insightful review that will definitely improve overall positioning of the paper. Thank you for updating the score.

---

### Official Review · Reviewer_7r9D · 2026-03-11

**Soundness:** 3
**Presentation:** 3
**Significance:** 3
**Originality:** 2
**Overall Recommendation:** 4
**Confidence:** 1

**Summary:**

The authors propose Continuous Depth Acceleration (CoDA), which uses a GRU module instead of continuous depth neural network (CDNN) layers. These GRU modules have PAC learnability guarantee, leveraging Koopman theory and Mori-Zwanzig formalism. This architecture bypasses ODE solvers and directly models iterations using GRU flows, providing compute and memory efficiency. This framework is applicable to Neural ODE and Liquid Time-Constant networks.

**Compliance With Llm Reviewing Policy:**

Affirmed.

**Final Justification:**

I retain my score because of the detailed response. I note that my confidence in this assessment is low, and I ask the Area Chair to weigh this accordingly.

**Key Questions For Authors:**

See the weakness part above.

**Limitations:**

They do not have a Impact Statement paragraph.

**Strengths And Weaknesses:**

Strength:

* They establish a theoretical guarantee about the learnability of such a GRU module.
* They explicitly analyze the theoretical FLOPs savings and practical inference time savings of the proposed architecture.
* They include results on diverse architectures to show the strength of their method.

Weakness:

* VF-Node seems to be a relatable baseline work that also tries to bypass ODE solvers and directly model a series of global integrals. The authors point out VF-Node is limited to NODE and cannot be applied to LTC Networks, but I do not understand the rationale behind this. What is the point that the proposed CoDA architecture makes it possible to apply to LTC Networks, but blocks VF-Node from being applied to LTC Networks?
* How is the VF-Node loss formulation different from CoDA? Do they have a different loss formulation, or do they only differ architecture-wise? It would be good if authors clarify how CoDA differs from state-of-the-art methods in a method-wise manner, and explain why they perform better. Is CoDA just a combination of VF-Node’s loss formulation and GRU Flow’s architecture design? If not, how does it differ beyond these state-of-the-art methods?
* Regarding Table 8, why does CoDA + GRUFlow perform better than NODE-VF? I wonder where this performance increase stems from. Is it because of better architecture, or a better loss formulation, or both? Also, how does the inference runtime compare between CoDA + GRUFlow and NODE-VF?
* Also in Table 8, I think the metric should be MSE (lower the better), not accuracy (higher the better).
* More broadly, I think it would be good if they have more comprehensive results comparing to NODE-VF more extensively beyond Table 8, as they seem to be a directly comparable state-of-the-art method.

---

> ### Author Rebuttal · Authors · 2026-03-30
>
> We thank Reviewer 7r9D for recognizing the theoretical guarantees, FLOPs analysis, and diverse evaluations. We address each concern below.
>
> ## Q1: Why can NODE-VF not be applied to LTC while CoDA can?
>
> NODE-VF's variational loss is derived from NODE-specific dynamics. The two have fundamentally different forward passes:
>
> **NODE:** dh/dt = tanh(W[i, h] + b)
>
> **LTC:** dh/dt = −h/(ρ/(1 + ρ·f_NN)) + f_NN
>
> NODE-VF's variational bounds assume the NODE form and cannot accommodate LTC's input-dependent time constants. CoDA operates via Koopman/Mori–Zwanzig theory, requiring only Lipschitz continuity — which both NODE and LTC satisfy.
>
> ## Q2: How does CoDA differ from NODE-VF and GRU Flow?
>
> CoDA is neither NODE-VF's loss nor GRU Flow's architecture. NODE-VF modifies the *loss* but retains the original architecture. CoDA replaces the *architecture* — substituting H nonlinear ODEs with P GRU cells + linear ODE + dense layer, using standard losses. The approaches are orthogonal. GRU Flow models the NODE integral directly with no Koopman/MZ basis and is limited to NODE, while CoDA's PAC guarantees extend to any Lipschitz-continuous CDLV.
>
> ## Q3: Why does CoDA outperform NODE-VF in Table 8, and inference runtime?
>
> The performance gain stems from architecture, not loss formulation — CoDA uses standard MSE loss, identical to the baseline. CoDA replaces H nonlinear ODEs with a compact P-dimensional linear ODE solved via matrix exponentiation, which provides two advantages: (1) the Koopman-based representation captures global dynamics in a lower-dimensional space, improving generalization, and (2) the linear ODE solver introduces no discretization error, unlike the iterative ODE solvers that NODE-VF's architecture still relies on at inference.
>
> For inference runtime on the VF-NODE-ICLR2025 dataset (Table 8), CoDA (P=16) achieves 0.0178 s/epoch compared to NODE-VF at 0.1894 s/epoch — a 10.6× speedup — while also achieving better MSE (2.72E-03 vs 3.54E-02). Even at P=1024, where CoDA achieves its best MSE (1.22E-03), inference takes 0.1225 s/epoch, still 1.5× faster than NODE-VF. This confirms that CoDA's architectural transformation provides both accuracy and speed advantages over NODE-VF's loss-based approach.
>
> ## Q4: Table 8 metric
>
> Correct — the metric is MSE (lower is better). The column header is mislabeled and will be fixed.
>
> ## Q5: More comprehensive NODE-VF comparison
>
> We have now run NODE-VF on all six benchmarks:
>
> **Training Time / Epoch (s):**
>
> | App. | CoDA | GRU Flow | NODE | LTC | NODE-VF |
> |---|---|---|---|---|---|
> | occupancy | 0.5 | 0.62 | 20.9 | 18.42 | 0.75 |
> | ozone | 0.1 | 0.2 | 2.62 | 2.57 | 0.26 |
> | person | 0.2 | 0.3 | 10.3 | 10.99 | 0.41 |
> | power | 4 | 5.5 | 272.6 | 210.09 | 5.82 |
> | har | 0.7 | 0.8 | 20.8 | 24.89 | 0.883 |
> | gesture | 0.1 | 0.143 | 3.25 | 2.51 | 0.16 |
>
> **Inference Time / Epoch (s):**
>
> | App. | CoDA | GRU Flow | NODE | LTC | NODE-VF |
> |---|---|---|---|---|---|
> | occupancy | 0.31 | 0.76 | 27.15 | 0.92 | 0.33 |
> | ozone | 0.03 | 0.04 | 2.57 | 1.21 | 0.026 |
> | person | 0.03 | 0.58 | 1.01 | 2.5 | 0.026 |
> | power | 0.16 | 0.07 | 0.36 | 0.001 | 0.059 |
> | har | 0.02 | 0.03 | 0.66 | 0.09 | 0.086 |
> | gesture | 0.0085 | 0.01 | 0.19 | 0.21 | 0.0066 |
>
> CoDA is 1.5–2.5× faster in training across all benchmarks because it eliminates the ODE solver entirely, while NODE-VF still requires iterative integration in the forward pass. For inference, results are mixed — CoDA is faster on some tasks (4× on har) while NODE-VF is faster on others (2.7× on power). Crucially, CoDA extends to LTC and other architectures, whereas NODE-VF cannot.
>
> ## Q6: Broader impact statement
>
> CoDA reduces the computational and energy costs of deploying continuous-depth neural networks, enabling their use on resource-constrained edge devices. However, by lowering the barrier to deployment in safety-critical domains such as healthcare and autonomous systems, there is a responsibility to ensure acceleration does not bypass thorough safety validation.

---

> > ### Author Rebuttal · Reviewer_7r9D · 2026-04-03
> >
> > Thank you for the detailed responses. I retain my score. I note that my confidence in this assessment is low, and I ask the Area Chair to weigh this accordingly.

---

### Official Review · Reviewer_XKTc · 2026-03-13

**Soundness:** 3
**Presentation:** 2
**Significance:** 3
**Originality:** 4
**Overall Recommendation:** 6
**Confidence:** 3

**Summary:**

The paper considers the problem of parametrizing as well as training continuous depth networks. The objective is to improve the computational efficiency as compared to other approaches. The paper shows that individual cells within continuous depth models can be mapped to simple linear dynamical systems that can be easily computed, with nonlinear maps going from one to the other representation. The evaluation is made in a range of tests. Theoretical results pertain to learnability guarantees.

**Compliance With Llm Reviewing Policy:**

Affirmed.

**Final Justification:**

Taking into account the extra experiments in the rebuttal, I raise my score

**Key Questions For Authors:**

- A high level question. What is the motivation for studying continuous depth models here? The paper improves upon other continuous depth models, but why not compare with standard discrete layer models? Are the other models still computationally easier than the CODA approach?

- The paper starts from a special case, as far as I can understand, of Neural ODEs, that involve separate cells with latent state variables and latent dynamics. But can we consider Neural ODE models where the input is mapped directly into a latent space, then Neural ODE dynamics evolve on this latent space until some terminal time, and then the final latent state is mapped to the output. Can this CODA method be used to improve such cases?

**Limitations:**

- Some technical notation errors. Looking at equation (2), it seems the calligraphic G function should have two inputs. but right before it says it is mapping just from R^H

- What is the decomposition into observable and implicit important? I missed this. Is this critical for the method to work? Actually, what is the definition of observable and implicit structure here and how is this enforced in the training?

- The matrix A is used in some places, but then it is defined within a proof. Please fix.

- Thereom 3.1., what is UAT?

- What is PESE in the text?

- Figure 4. If I understand, the curves are made by fitting into the shown data points. If this is true, this is not appropriate. These are too few data points to understand the trend as P changes.

**Strengths And Weaknesses:**

- The idea of using Koopman theory to simplify continuous depth neural nets is novel and interesting -- probably a matter of time before someone else implements this. I am more aware of the opposite direction, using deep neural nets to learn Koopman models of dynamical systems, which has received considerable attention. It is likely that the present paper will receive similar attention.

- The numerical experiments are thorough and supportive.

- Even though I am not an expert in parts of the topics, GRUs and LTVs, I found the presentation sufficient, but I note some technical points later.

---

> ### Author Rebuttal · Authors · 2026-03-30
>
> We thank Reviewer XKTc for the positive assessment and thorough technical feedback. We address each point below.
>
> ## Q1: Why study continuous-depth models? Why not compare with standard discrete layer models?
>
> Yes, standard discrete models (e.g., LSTM, GRU) are computationally easier than unaccelerated CDNNs. Tables 3–4 confirm this. However, CDNNs often achieve superior accuracy on dynamical systems tasks (e.g., NODE and LTC outperform LSTM on person activity, HAR, and gesture). CoDA makes CDNNs computationally competitive with discrete models while retaining their accuracy advantages — after applying CoDA, the accelerated CDNN matches or beats LSTM in both speed and accuracy across most benchmarks.
>
> We have also applied CoDA to four discrete architectures — CoLA, RWKV, MoEUT, and Huginn — achieving 2.3–2.9× speedups, demonstrating that CoDA is beneficial even beyond the continuous-depth setting.
>
> |Arch.|H→P|Seq32|Seq64|Seq128|Seq256|
> |---|---|---:|---:|---:|---:|
> |CoLA|768→16|2.72×|2.71×|2.21×|1.64×|
> |RWKV|768→16|2.59×|2.73×|2.33×|1.61×|
> |MoEUT|1024→64|2.69×|2.56×|2.43×|0.87×|
> |Huginn|5280→256|2.91×|2.79×|2.38×|1.70×|
>
> ## Q2: Can CoDA be applied to encoder → Neural ODE → decoder architectures?
>
> Yes. CoDA's H-P transformation applies to the Neural ODE dynamics block: the H-dimensional nonlinear ODE is replaced by P GRU cells + a linear ODE + a dense layer, while encoder and decoder remain unchanged. Table 7 (MNIST, NODE H=512, P=16) demonstrates precisely this — CoDA achieves 92% test accuracy with 3× inference speedup. We will clarify this more explicitly in the revision.
>
> ## Q3: Notation error in Equation (2)
>
> Thank you. G should take two inputs: G(h(t), i(t)), mapping from R^H × R^N → R^P. We will fix this.
>
> ## Q4: Why is the observable/implicit decomposition important?
>
> The Koopman operator is potentially infinite-dimensional, which cannot be tractably represented. The Mori–Zwanzig formalism provides a way out: only a finite set of observable dynamics needs explicit modeling. All other dynamics can be approximated as implicit effects modeled by neural networks as universal approximators.
>
> This maps directly to CoDA's architecture (Figure 2): Block 1 models finite observable dynamics (Part 1, Equation 8), Block 2 models implicit dynamics via learned convolution (Part 2, Theorem 3.1), and the linear ODE solver integrates the combined observables. The decomposition is not explicitly enforced during training — the weight constraints in Table 1 structurally partition the GRU into observable and implicit blocks, and the network learns the decomposition through backpropagation within these constraints.
>
> ## Q5: Matrix A used before being defined
>
> We will move the definition of A — a (P+N) × (P+N) matrix concatenating an identity matrix P × P and weight matrix W_h of size N × N — into the main text at first use.
>
> ## Q6: What is UAT?
>
> UAT stands for Universal Approximation Theorem. We will spell it out at first use.
>
> ## Q7: What is PESE?
>
> Initially we named our technique PESE, but later changed to CoDA. Several instances were not updated during manuscript preparation. We will correct this.
>
> ## Q8: Figure 4 — too few data points for trend fitting
>
> We agree. We will remove the fitted curves and connect data points directly. The points at P ∈ {16, 64, 144, 256, 576, 1024} already show a clear monotonic relationship — test loss decreases and training time increases as P grows — consistent with the MZ formulation's prediction that larger P models more dynamics explicitly.
>
> ## Q9: Equation 8 vs Equation 19 discrepancy
>
> Equation 8 is a simplified form; Equation 19 is the full derivation including the residual term G_I(0). Under the steady-state assumption (appendix), this residual vanishes for long sequences, reducing Equation 19 to Equation 8. We will state this relationship and assumption explicitly in the main text.

---

> > ### Author Rebuttal · Reviewer_XKTc · 2026-04-02
> >
> > I appreciate the extra experiments

---

> > > ### Author Response · Authors · 2026-04-02
> > >
> > > Thank you for your positive comments and updating the score.

---

### Official Review · Reviewer_jgR1 · 2026-03-18

**Soundness:** 4
**Presentation:** 4
**Significance:** 3
**Originality:** 4
**Overall Recommendation:** 5
**Confidence:** 4

**Summary:**

This paper introduces a new sequence model based on a continuous time GRU. Theoretical results demonstrate the generality of the formulation. The proposed architecture is primarily utilized as a drop-in replacement into the Liquid Foundation Model.  Experiments are implemented on mobile GPUs and FPGAs and runtime and power usage is compared. The proposed architecture out performs comparisons particularly along the efficiency axis in both training and inference on edge hardware.

**Compliance With Llm Reviewing Policy:**

Affirmed.

**Key Questions For Authors:**

- What are the parameter counts of the different models?
- Can the authors attributed exactly where does the speed up and improvement for the proposed layer come from? It is reported that the layers take up less RAM on certain architectures: is it entirely from reducing parameter count?
- What happens when you scale the proposed layer to be larger to keep costs fixed? Does the accuracy metrics improve, or is this method only useful for efficiency?
- Is it possible to include a result that uses the CoDA layer as an RNN or encoder-decoder RNN, trained from scratch, to compare it to the LFM version?
- Are the models trained from scratch, or do you start from a checkpoint and then do the swap in Figure 2?
- The appendix switched to italics midway through

**Limitations:**

The paper is missing the required broader impact statement. The authors will need to address this.

**Strengths And Weaknesses:**

This is a very detailed paper. In spans both theoretical backings, implementation details, and extensive benchmarking of the proposed method. The results are well presented and support the improvements of the proposed architecture. Experiments wide range of serious benchmark tasks and modalities are reported. The detailed breakdown of flops/memory movement/power consumption is a major strength of the paper, as is demonstrating FPGA implementation.

One particular weakness is that the paper does not vary the size of architecture and only reports one instantiation of it. The proposed architecture is only explored inside the LFM: Is the architecture applicable beyond the LFM? Can it be used as a normal recurrent model in isolation? It does not Reporting the performance with different hyperparameters and configurations is good for architecture proposals.

---

> ### Author Rebuttal · Authors · 2026-03-30
>
> We thank Reviewer jgR1 for the thorough, constructive review and for recognizing the theoretical backing, benchmarking, and hardware analysis as strengths.
>
> ## Q1: Only one architecture size is reported
> Figure 4 and Table 8 already vary CoDA layer size with P ∈ {16,64,144,256,576,1024}, and Table 7 uses H = 512 (vs. H = 1024 elsewhere).
>
> **New results:** We evaluated CoDA on four additional architectures with different hidden dimensions and parameter counts:
>
> |Arch.|Params|H|P|
> |---|---:|---:|---:|
> |CoLA|~191M|768|16|
> |RWKV|~169M|768|16|
> |MoEUT|~117M|1024|64|
> |Huginn|~153M|5280|256|
>
> *Table R1. New architecture configurations.*
>
> Together with the main paper (H = 512 in Table 7, H = 1024 in Tables 2–4, LFM2 at 1,170M), CoDA is now evaluated across H ∈ {512,768,1024,5280}, spanning a 10× hidden-dimension range and models from ~117M to 1,170M parameters. Speedups are in Table R2.
>
> ## Q2: Is CoDA applicable beyond the LFM? standalone recurrent model
> Yes. Lemma 3.1 and Theorem 3.1 apply to any CDLV layer satisfying Lipschitz continuity, not just LFM. Tables 3–4 already show six standalone LTC-NN applications; Table 7 shows NODE (MNIST), and Table 8 shows VF-NODE.
>
> ### New results: generalization to RNN architectures
> We further evaluated CoDA on CoLA, RWKV, MoEUT, and Huginn. These show the H→P transformation generalizes beyond continuous-depth models.
> Sequence length (Seq)
> |Arch.|H→P|Seq32|Seq64|Seq128|Seq256|Test Loss Base (Seq256)|Test Loss CoDA(Seq256)|
> |---|---|---:|---:|---:|---:|---|---|
> |CoLA|768→16|2.72×|2.71×|2.21×|1.64×|0.0810|0.0751|
> |RWKV|768→16|2.59×|2.73×|2.33×|1.61×|2.9642|2.8783|
> |MoEUT|1024→64|2.69×|2.56×|2.43×|0.87×|6.4844|6.2965|
> |Huginn|5280→256|2.91×|2.79×|2.38×|1.70×|2.5177|2.5|
>
> *Table R2. CoDA speedup across four non-CDNN architectures.*
>
> CoDA delivers 2.3–2.9× speedups at seq. lengths 32–128 across all four architectures. At seq. 256, speedups drop due to matrix-exponential memory overhead; three still retain 1.6–1.7× gains. MoEUT slows at seq. 256 (0.87×), likely because dense reconstruction interacts poorly with MoE routing and adds memory traffic. We are investigating this and will report test loss with timing in the camera-ready version. Huginn (H = 5280) achieves the best short-sequence speedup, suggesting favorable scaling with hidden dimension.
>
> **Training from scratch:** In Tables 3, 4, 7, and 8, all CoDA models are trained from scratch. Only the LFM2-1.2B ablation in Table 2 uses distillation from LFM2-7B.
>
> ## Q3: Parameter counts of different models
> We provide parameter counts for all models:
>
> |Model|Params|
> |---|---|
> |LTC-NN (H = 1024)|~2.1M/CDLV layer|
> |LFM2-1.2B|1,170M|
> |CoLA Open AI 2019 (H = 768)|~191M|
> |RWKV Peng EMNLP 23 (H = 768)|~169M|
> |MoEUT Csordas Neurips 24 (H = 1024)|~117M|
> |Huginn Geiping Neurips 25 (H = 5280)|~153M|
> |NODE (H = 512)|~524K/layer|
> |GRUFlow|~786K|
> |LSTM (H = 128)|~264K|
> |CoDA (P = 16)|~68K|
> |CoDA (P = 64)|~200K|
> |CoDA (P = 1024)|~3.2M|
>
> *Table R3. Parameter counts for all compared models.*
>
> ## Q4: Source of speedup?
> It comes from three sources:
>
> **1. Eliminating iterative ODE solving:** Standard CDLV layers require N_steps sequential nonlinear ODE integrations with complexity O(N_steps·H²). CoDA replaces this with a single matrix exponentiation at O(P²), removing the sequential bottleneck.
>
> **2. Reduced dimensionality (P ≪ H):** The intermediate state is P-dimensional, reducing compute and memory traffic. Table 5 shows FLOPs drop from 95.66 to 10.53 GFLOPs and I/O from 0.031 to 0.012 GB/token.
>
> **3. Hardware-friendly computation:** Matrix exponentiation maps well to tensor cores and systolic arrays, enabling fusion and better cache use. Table 11 shows improved L1 hit rate, and Table 6 shows FPGA RAM drops from 495 MB to 214 MB.
>
> ## Q5: Scale P larger to keep costs fixed?
> Figure 4 addresses this directly. As P increases from 16 to 1024, test loss decreases while training time increases. Table 8 shows CoDA (P = 1024) achieves 1.22E-03 vs. 2.72E-03 for P = 16, at ~7× higher training time (0.1225 vs. 0.0178 s/epoch). Even at P = 1024, CoDA remains faster than GRUFlow (0.3691 s/epoch) and NODE-VF (0.1894 s/epoch) while achieving better accuracy.
>
> ## Q6: Can CoDA be used as an RNN trained from scratch?
> Yes. Tables 3, 4, 7, and 8 train CoDA from scratch as a standalone module. The CoDA block does not require a pretrained CDNN checkpoint. Only Table 2 uses distillation, and even there the CoDA layers are freshly initialized.
>
> ## Q7: Broader impact
> We will add a broader impact in revision. CoDA enables energy-efficient edge AI deployment, reduces training/deployment carbon cost, and broadens access on memory-constrained hardware. Although CoDA can preserve accuracy, strict safety validation is required before deployment because CoDA gives a learnability guarantee, but insufficient training can still reduce accuracy.

---

> > ### Author Rebuttal · Reviewer_jgR1 · 2026-04-03
> >
> > I thank the authors for the response and clarifications. I am still very supportive of acceptance. My one recommendation is to add more explanation of the end-to-end architecture set up for the CoDA for each of the problems (i.e. what are the input and output layers of the models and what are the loss functions) for the cases where CoDA is used standalone and the cases where it is plugged into the LFM. One or two clarifying lines in the abstract and experimentation section to state that it is used standalone and in the LFM would help.

---

### Decision · Program_Chairs · 2026-04-30

**Decision:**

Accept (regular)

**Comment:**

The paper proposes a drop-in replacement for continuous-depth latent variable layers (e.g. neural ODE layers) that require multiple nonlinear ODE solves.
The replacement is motivated by Koopman operator theory, shown to be equivalent in terms of PAC-learnability, and requires only a single linear ODE solve.
Empirically, the authors show that their drop-in replacement, called CoDA, preserves accuracy while enabling training and inference speedups.

Overall, the reviewers appreciated the combination of theoretical contribution, implementation details, and extensive experiments.
There were requests to extend the experiments to RNN architectures and run additional baselines on all benchmarks, which the authors addressed.
The authors also further clarified the sources of speedup in their approach (reducing number of ODE solves, and the ODE dimension) and provided the required impact statement that was missing in the original submission.
Post-rebuttal and -discussion, there remain no major weaknesses.
The paper seems to take an important step towards reducing the computational cost in continuous depth latent variable layers with a mathematically sound argument, and sufficient empirical evidence.
Therefore, I recommend acceptance.

There were some typesetting issues which I believe should be fixed.
I highly recommend the authors increase the font size for their mathematical equations, as they are hard to read in the current version.
It would also be good to add more intuition to the proof sketches, which are a bit mechanical.
There also seems to be a mistake in the dimensions of self.A in Figure 3.